# JMJD3 acts in tandem with KLF4 to facilitate reprogramming to pluripotency

Yinghua Huang [1,2,17], Hui Zhang[3,17], Lulu Wang [1,2], Chuanqing Tang [1,2], Xiaogan Qin[1,2], Xinyu Wu [1,2,4], Meifang Pan[1,2,4], Yujia Tang[1,2], Zhongzhou Yang[1], Isaac A. Babarinde[5], Runxia Lin[1,2,4], Guanyu Ji [6], Yiwei Lai [1,7], Xueting Xu [1,2,8], Jianbin Su[1,2], Xue Wen[9], Takashi Satoh[10], Tanveer Ahmed [1,2], Vikas Malik [1,7], Carl Ward [1,7], Giacomo Volpe [1,7], Lin Guo [1], Jinlong Chen[1], Li Sun[5], Yingying Li[3], Xiaofen Huang[1], Xichen Bao [1,3,11], Fei Gao[6,12], Baohua Liu [13], Hui Zheng [1,3,11], Ralf Jauch [14], Liangxue Lai[1,3,11], Guangjin Pan [1,3,11], Jiekai Chen [1,3,11], Giuseppe Testa [15], Shizuo Akira[10], Jifan Hu [9], Duanqing Pei [1,3], Andrew P. Hutchins [5], Miguel A. Esteban [1,7,11,16✉] & Baoming Qin [1,2,3,11✉]

The interplay between the Yamanaka factors (OCT4, SOX2, KLF4 and c-MYC) and transcriptional/epigenetic co-regulators in somatic cell reprogramming is incompletely understood. Here, we demonstrate that the histone H3 lysine 27 trimethylation (H3K27me3) demethylase JMJD3 plays conflicting roles in mouse reprogramming. On one side, JMJD3 induces the pro-senescence factor *Ink4a* and degrades the pluripotency regulator PHF20 in a reprogramming factor-independent manner. On the other side, JMJD3 is specifically recruited by KLF4 to reduce H3K27me3 at both enhancers and promoters of epithelial and pluripotency genes. JMJD3 also promotes enhancer-promoter looping through the cohesin loading factor NIPBL and ultimately transcriptional elongation. This competition of forces can be shifted towards improved reprogramming by using early passage fibroblasts or boosting JMJD3's catalytic activity with vitamin C. Our work, thus, establishes a multifaceted role for JMJD3, placing it as a key partner of KLF4 and a scaffold that assists chromatin interactions and activates gene transcription.

[1] Chinese Academy of Sciences (CAS) Key Laboratory of Regenerative Biology and Guangdong Provincial Key Laboratory of Stem Cells and Regenerative Medicine, Guangzhou Institutes of Biomedicine and Health (GIBH), CAS, 510530 Guangzhou, China. [2] Laboratory of Metabolism and Cell Fate, GIBH, CAS, 510530 Guangzhou, China. [3] Bioland Laboratory (Guangzhou Regenerative Medicine and Health Guangdong Laboratory), 510005 Guangzhou, China. [4] University of Chinese Academy of Sciences, 100049 Beijing, China. [5] Department of Biology, Southern University of Science and Technology, 518055 Shenzhen, China. [6] Science and Technology Department, E-GENE, 518118 Shenzhen, China. [7] Laboratory of Integrative Biology, GIBH, CAS, 510530 Guangzhou, China. [8] School of Life Sciences, University of Science and Technology of China, 230027 Hefei, China. [9] Stem Cell and Cancer Center, First Affiliated Hospital, Jilin University, 130061 Changchun, China. [10] Department of Host Defense, Research Institute for Microbial Diseases, Osaka University, Osaka 565-0871, Japan. [11] Joint School of Life Sciences, GIBH and Guangzhou Medical University, 511436 Guangzhou, China. [12] Agricultural Genomics Institute at Shenzhen, Chinese Academy of Agricultural Sciences, 518120 Shenzhen, China. [13] Health Science Center, Shenzhen University, 518060 Shenzhen, China. [14] School of Biomedical Sciences, Li Ka Shing Faculty of Medicine, The University of Hong Kong, Hong Kong, SAR, China. [15] Department of Experimental Oncology, European Institute of Oncology, Milan 20139, Italy. [16] Institute for Stem Cells and Regeneration, CAS, 100101 Beijing, China. [17] These authors contributed equally: Yinghua Huang, Hui Zhang. ✉email: miguel@gibh.ac.cn; qin_baoming@gibh.ac.cn

During the reprogramming of somatic cells to induced pluripotent stem cells (iPSCs) with exogenous factors, there is a comprehensive epigenetic transformation to establish an embryonic stem cell (ESC)-like transcriptional program[1]. To induce this epigenetic reprogramming, the Yamanaka factors (OCT4, SOX2, KLF4, and c-MYC; OSKM) recruit multiple epigenetic enzymes and other co-regulators to target loci[2]. Manipulating the activity or expression levels of these factors has a substantial impact on the efficiency and kinetics of iPSC generation[3]. A remarkable example is the addition of vitamin C (Vc) to reprogramming media[4], which boosts the process by enhancing the function of histone demethylases and TET DNA hydroxylases[5]. Yet, despite these and other discoveries, our understanding of the spatial–temporal cooperation between OSKM and epigenetic factors in reprogramming remains fragmentary and, in fact, some of the reported observations are seemingly contradictory[2,6,7]. A major source of confusion is that some regulators work differently in various reprogramming contexts and/or reprogramming phases.

H3K27me3 is a major silencing mechanism controlling development and stem cell differentiation. It is deposited by the polycomb repressive complex 2 (PRC2)[8] and removed by two histone demethylases: JMJD3 (also known as KDM6B) and UTX (KDM6A)[9]. PRC2 and JMJD3/UTX are dispensable for pluripotency maintenance in ESCs[10–12]. Yet, while PRC2 and UTX are necessary for efficient reprogramming[13,14], JMJD3 is thought to be detrimental[15]. Of note, JMJD3 and UTX share a high degree of similarity in their catalytic domains but often have different, sometimes opposing, functions[16].

Here, we demonstrate that, JMJD3 is a potent activator of epithelial and pluripotency genes in mouse reprogramming with OSKM. KLF4 mediates the recruitment of JMJD3 to the enhancers and promoters of these genes, where both factors cooperate with p300, cohesin, mediator, and other co-regulators, in particular the cohesin loading factor NIPBL. This promotes a switch from H3K27me3 to H3K27 acetylation (H3K27ac), enhancer-promoter looping and triggers productive transcriptional elongation at target loci. Simultaneously, JMJD3 induces H3K27me3 demethylation at the pro-senescence Ink4a/Arf locus, and degradation of PHF20, a component of the histone acetyltransferase MOF–NSL complex involved in pluripotency regulation[15], with both effects being independent of KLF4 or reprogramming. When basal cell senescence is high, the negative force of JMJD3 dominates, whereas in young fibroblasts JMJD3 enhances reprogramming and this is potentiated by Vc. Notably, we also show that JMJD3 not only promotes iPSC generation from fibroblasts and incompletely reprogrammed iPSCs (pre-iPSCs)[17], but also facilitates the KLF4-mediated mesenchymal-to-epithelial transition (MET) and the primed-to-naïve pluripotency transition[18,19].

Our results, thus, establish a new picture for JMJD3 and KLF4 in multiple cell fate conversions, which has implications for understanding the complex roles of these two factors in normal physiology and disease.

## Results

**Dual effects of JMJD3 on somatic cell reprogramming.** The function of both JMJD3 and UTX is to reduce the levels of H3K27me3, a highly dynamic epigenetic mark in reprogramming[20]. Moreover, mRNA expression of both enzymes measured by quantitative PCR with reverse transcription (RT-qPCR) is higher in ESCs than MEFs, and increases progressively during reprogramming (Supplementary Fig. 1a). To study the role of JMJD3 in reprogramming in more detail, we overexpressed JMJD3 (Supplementary Fig. 1b) in Oct4-GFP reporter (OG2)

MEFs transduced with OSKM in standard (serum + LIF) culture conditions[21]. Stress-induced JMJD3 activates pro-senescence Ink4a/Arf (Cdkn2a) in different cell types including MEFs[22], and cell senescence is detrimental for reprogramming[23]. Thus, we conceived that the senescence state of MEFs may influence JMJD3's effect in reprogramming. As shown in Fig. 1a, Ink4a expression increases and cell proliferation decreases during routine passaging of MEFs. However, endogenous Jmjd3 or Utx did not change (Fig. 1b), suggesting that the induction of Ink4a by serial passaging is unrelated. Accordingly, we conducted reprogramming in both early (passage 2: P2) and late (P4) passage MEFs, and also tested the effect of adding Vc[4] because it boosts the catalytic activity of Jumonji C (JmjC)-domain-containing enzymes including JMJD3[5].

As expected, exogenous JMJD3 increased the expression of Ink4a and decreased proliferation of reprogramming cells (Fig. 1c and Supplementary Fig. 1c). In agreement with a previous report[15], JMJD3 reduced the number of alkaline phosphatase positive (AP+) colonies in both P2 and P4 MEFs with or without Vc (Fig. 1d, e). But AP is a marker of the early phase of reprogramming and, interestingly, JMJD3 simultaneously increased the number of GFP+ colonies in P2 MEFs, especially with Vc, though it did the opposite in P4 MEFs (Fig. 1d, e). The synergistic effect of Vc and JMJD3 in P2 MEF reprogramming was not mediated by an attenuation of cell senescence, as Ink4a levels were not affected by Vc (Fig. 1c). The stringency of OG2 GFP+ colony quantification as readout for reprogramming efficiency could be verified using Dppa5a-tdTomato/OG2 dual-reporter MEFs[24] and flow cytometry (Supplementary Fig. 1d). Importantly, iPSCs generated with JMJD3 overexpression (OE) and Vc in P2 MEFs were fully pluripotent, as characterized by standard procedures (Supplementary Fig. 1e–h). We overexpressed UTX as a control (Supplementary Fig. 1b), observing, as reported[14], no effect on AP+ or GFP+ colony formation in P2 MEFs with or without Vc (Supplementary Fig. 1i). In addition, exogenous JMJD3 achieved similar results using early (P1) and late (P3) passage tail tip fibroblasts (TTFs; Fig. 1f). Likewise, exogenous JMJD3 enhanced GFP+ colony formation in P2 MEFs in iSF1[25], a high-efficiency serum-free reprogramming medium containing Vc (Supplementary Fig. 1j). These experiments showed that the synergistic effect of JMJD3 and Vc in reprogramming does not depend on the choice of basal medium or fibroblast type.

Because JMJD3 reduces AP+ but enhances GFP+ colonies in P2 MEFs, we postulated that it could be inhibiting the early stage of reprogramming and promoting the late stage. To test this, we took advantage of the cell surface markers, CD44 and ICAM1 (or CD54), which are expressed in different combinations at different stages of reprogramming[26]. Flow cytometry confirmed that JMJD3 OE slows down the disappearance of the somatic state (ICAM1+CD44+ cells) at day 5 while accelerating the appearance of the pluripotent state (ICAM1+CD44- cells) at day 10 (Supplementary Fig. 1k). Along the same line of thought, we demonstrated that JMJD3 promotes the conversion of incompletely reprogrammed colonies (pre-iPSCs, AP+ and GFP−) to bona fide iPSCs[4] with no change in AP+ colony number (Fig. 1g). This is likely because pre-iPSCs have already gone through the initial phase (containing the senescence barrier) of reprogramming.

Besides activating Ink4a/Arf, JMJD3 impairs reprogramming by inducing the degradation of PHF20[15]. Western blotting confirmed that overexpressing JMJD3 in P2 MEFs reduces PHF20 in addition to enhancing p16INK4A, also indicating that both effects are reprogramming independent. Likewise, we observed that Vc does not prevent JMJD3-induced degradation of PHF20 in either P2 MEFs or OSKM reprogramming (Supplementary

Fig. 2a). Accordingly, simultaneous overexpression of PHF20 and JMJD3 synergistically enhanced reprogramming (Supplementary Fig. 2b, c).

We concluded that JMJD3 has dual effects in reprogramming, derailing it by inducing *Ink4a/Arf* expression and the degradation of PHF20, but also promoting through a yet unclear mechanism

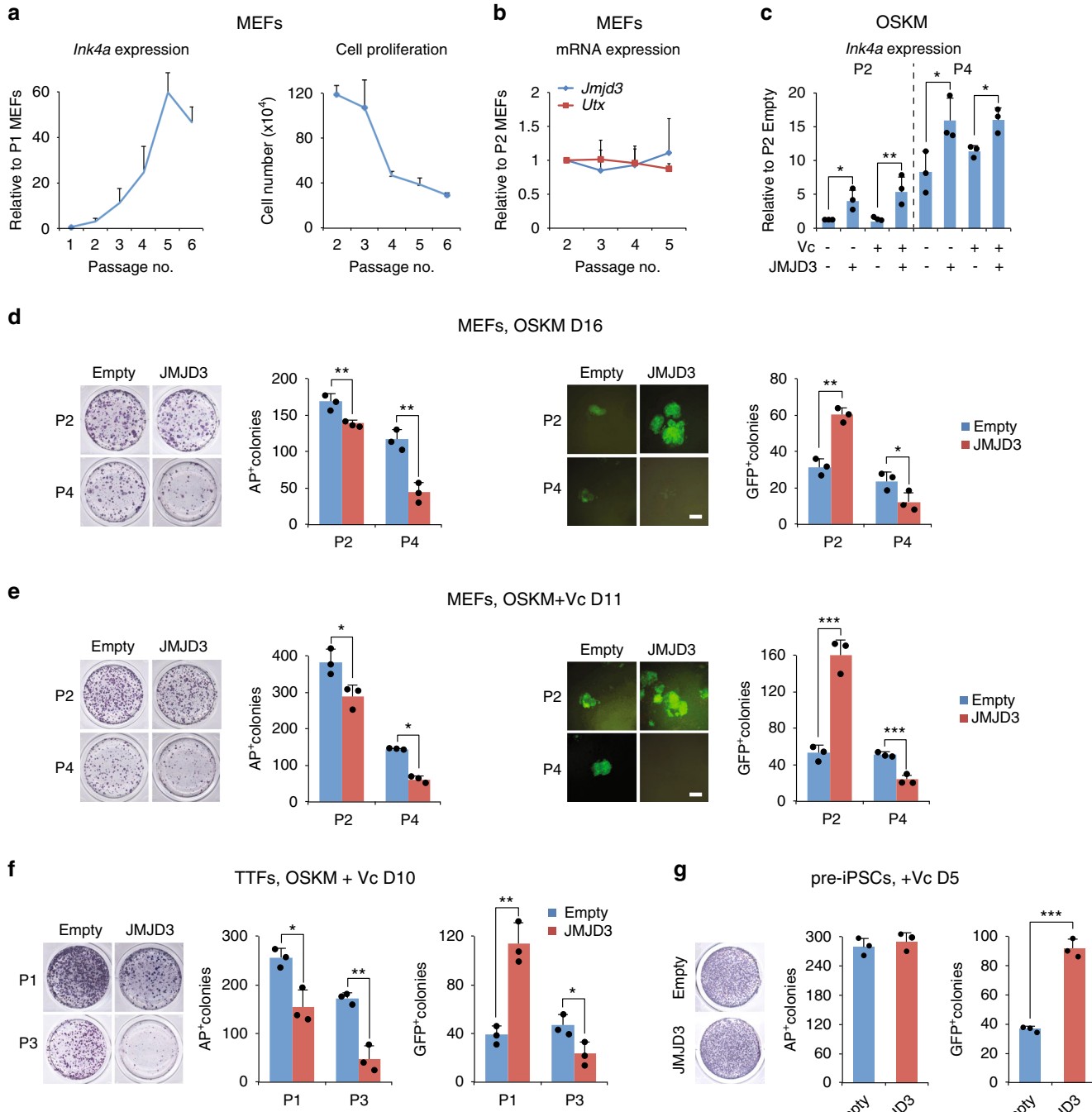

**Fig. 1 The senescence state of fibroblasts determines JMJD3's effect in reprogramming. a** Correlation between *Ink4a* induction and cell proliferation in a serial passaging of MEFs. $2 \times 10^5$ MEFs per well of a six-well plate were seeded and cell number was counted at day 3 before each passaging. **b** RT-qPCR for *Jmjd3* and *Utx* in a serial passaging of MEFs. **c** RT-qPCR for *Ink4a* in P2 and P4 MEFs transduced with OSKM and empty vector (Empty) or JMJD3 in medium with or without Vc. **d, e** Images and numbers of AP+ colonies (left panel) and *Oct4*-GFP+ colonies (right panel) at day 16 (without Vc, **d**) or day 11 (with Vc, **e**) in P2 and P4 MEFs transduced with OSKM and Empty or JMJD3. D day. Scale bar, 200 μm. **f** AP staining images and colony numbers (AP+ and GFP+) at day 10 in P1 and P3 TTFs transduced with OSKM and Empty or JMJD3 in medium with Vc. **g** AP staining images and colony numbers (AP+ and GFP+) at day 5 in pre-iPSCs transduced with Empty or JMJD3. Error bars represent the s.e.m. Data are presented as mean ± s.e.m. from $n = 3$ **a**–**g** biologically independent experiments. Statistical analyses were performed using a two-tailed unpaired Student's *t*-test **d**–**g**, or one-way analysis of variance (ANOVA) followed by a Holm–Sidak multiple comparison test **c** (*$P < 0.05$; **$P < 0.01$; ***$P < 0.001$). *P* values: 0.0252, 0.0086, 0.0111, 0.0493 **c**; 0.0095, 0.0031, 0.0012, 0.042 **d**; 0.0267, $7.98 \times 10^{-5}$, 0.0005, 0.0043 **e**; 0.0119, 0.0018, 0.0024, 0.0344 **f**; 0.0001 **g**. Source data are provided as a Source Data file.

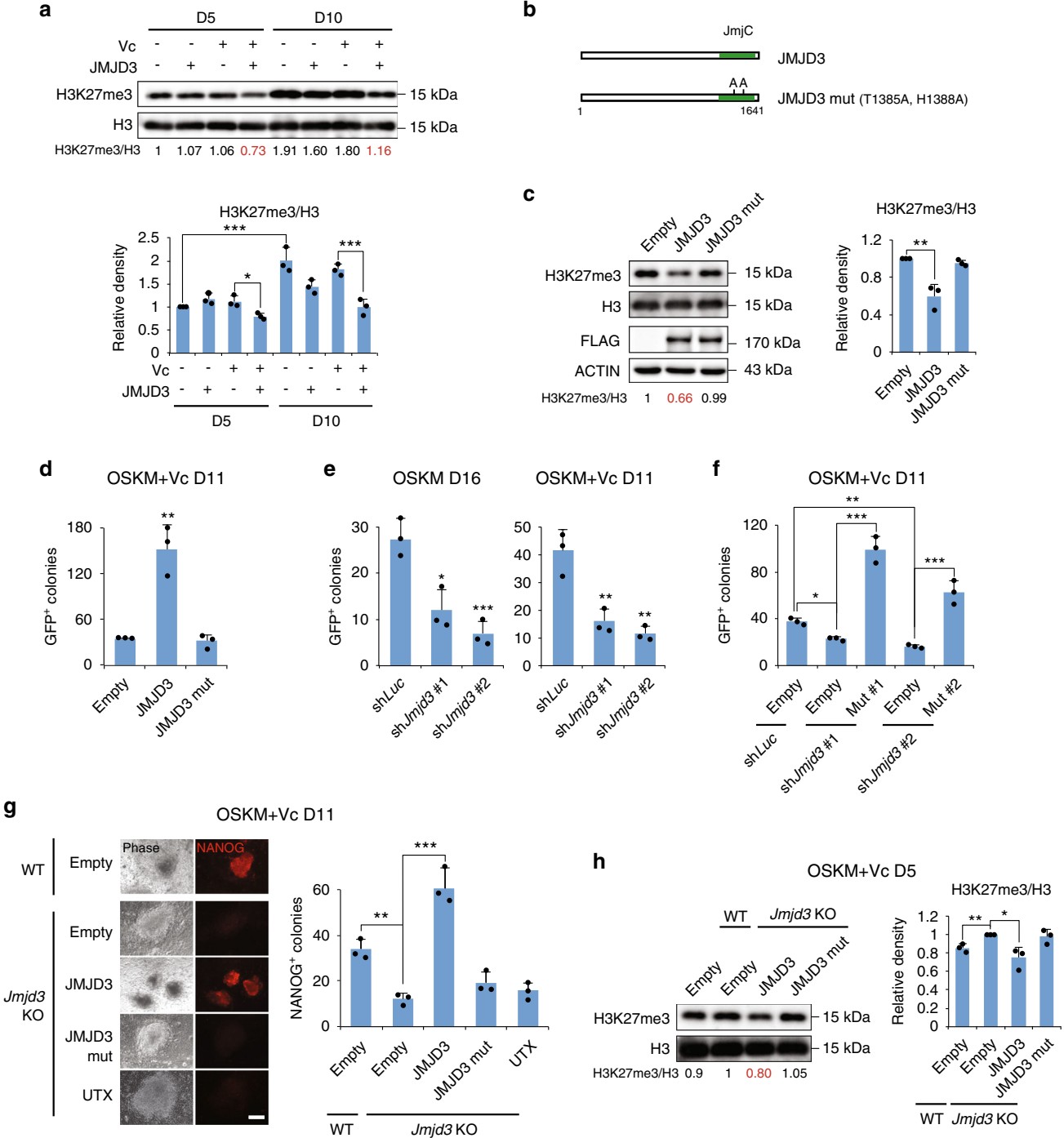

that can be further enhanced by adding Vc. The balance between both effects determines the ultimate outcome.

**JMJD3 boosts reprogramming in a catalytic-dependent manner.** To study how JMJD3 OE enhances reprogramming in early passage fibroblasts, we first analyzed H3K27me3 levels by western blotting and found a global increase in iPSCs and during reprogramming (Fig. 2a and Supplementary Fig. 3a). JMJD3 significantly reduced H3K27me3 during reprogramming, especially in the presence of Vc (Fig. 2a and Supplementary Fig. 3b). We then mutated amino acids Thr1385 and His1388 to Ala in the JmjC domain of JMJD3 to produce a catalytically inactive mutant[27] (Fig. 2b). These mutations abolished the effect of exogenous JMJD3 on H3K27me3 levels in reprogramming with Vc (Fig. 2c)

and also on reprogramming efficiency (Fig. 2d). Therefore, JMJD3 catalytic activity is needed for its enhancing effect in early passage fibroblast reprogramming.

Next, we studied whether endogenous JMJD3 behaves similarly, and is necessary for efficient reprogramming by controlling H3K27me3 levels. *Jmjd3* knockdown in P2 MEFs (Supplementary Fig. 4a) impaired GFP⁺ colony formation in serum + LIF with or without Vc and in iSF1 medium (Fig. 2e and Supplementary Fig. 4b). Likewise, *Jmjd3* knockdown inhibited pre-iPSC reprogramming to full iPSCs (Supplementary Fig. 4c). As previously reported[14], knocking down *Utx* also reduced GFP⁺ colony formation (Supplementary Fig. 4d, e). We confirmed the specificity of the shRNAs by constructing shRNA-resistant JMJD3 and UTX mutants (Supplementary Fig. 4f), which showed

**Fig. 2 JMJD3 enhances reprogramming efficiency in a catalytic-dependent manner. a** Western blotting for H3K27me3 in the indicated OSKM-reprogrammed P2 MEFs and quantification normalized to H3 (upper panel), and the statistics for the quantification (lower panel). **b** Schematic depiction of JMJD3 and a catalytic mutant JMJD3 (JMJD3 mut). **c** Western blotting for H3K27me3 and FLAG (exogenous FLAG-tagged JMJD3 constructs) in OSKM-reprogrammed P2 MEFs at day 5 and H3K27me3 quantification normalized to H3 (left panel) and statistics for the quantification (right panel). **d** Number of GFP⁺ colonies at day 11 in OSKM-reprogrammed P2 MEFs with the indicated JMJD3 constructs. **e** Number of GFP⁺ colonies at days 11 (with Vc) and 16 (without Vc) in OSKM-reprogrammed P2 MEFs with shRNAs against firefly *Luciferase* (sh*Luc*) or *Jmjd3* (sh*Jmjd3*; two independent shRNAs). **f** Number of GFP⁺ colonies at day 11 in OSKM-reprogrammed P2 MEFs with sh*Luc* or sh*Jmjd3*, together with the indicated JMJD3 rescue mutants or Empty. **g** Phase contrast and NANOG immunofluorescent images of typical colonies (left panel) and number of NANOG⁺ colonies in OSKM-reprogrammed WT and *Jmjd3* KO P2 MEFs with the indicated constructs (right panel) at day 11. Scale bars, 100 μm. **h** Western blotting for H3K27me3 in OSKM-reprogrammed WT and *Jmjd3* KO P2 MEFs with the indicated constructs at day 5, H3K27me3 quantification normalized to H3 (left panel) and statistics for the quantification (right panel). Error bars represent the s.e.m. Data are presented as mean ± s.e.m. from $n = 3$ **a**, **d–h** biologically independent experiments. Statistical analyses were performed using a two-tailed unpaired Student's $t$-test **c–e**, **g**, **h** or one-way ANOVA followed by a Holm–Sidak multiple comparison test **a**, **f** (*$P < 0.05$; **$P < 0.01$; ***$P < 0.001$). $P$ values: 0.0198 (D5 + Vc − JMJD3 vs. D5 + Vc + JMJD3), 0.0001 (D5 − Vc − JMJD3 vs. D10 − Vc − JMJD3), 0.0001 (D10 + Vc − JMJD3 vs. D10 + Vc + JMJD3) **a**; 0.0059 **c**; 0.0031 **d**; 0.0135, 0.0009, 0.0048, 0.0023 **e**; 0.0255 (sh*Luc* + Empty vs. sh*Jmjd3* #1 + Empty), 0.0001 (sh*Jmjd3* #1 + Empty vs. sh*Jmjd3* #1 + Mut #1), 0.0065 (sh*Luc* + Empty vs. sh*Jmjd3* #2 + Empty), 0.0001 (sh*Jmjd3* #2 + Empty vs. sh*Jmjd3* #2 + Mut #2) **f**; 0.0018, 0.0009 **g**; 0.0053, 0.0189 **h**. Source data are provided as a Source Data file.

a rescue effect on reprogramming efficiency (Fig. 2f and Supplementary Fig. 4g). The need for endogenous JMJD3 in the reprogramming of early passage MEFs with Vc was also validated using littermate wild-type (WT) and *Jmjd3* knockout (KO) P2 MEFs[28] (Supplementary Fig. 4h). As these MEFs do not contain an OG2, we used immunofluorescence staining for NANOG to evaluate reprogramming efficiency. NANOG staining in OG2 MEF reprogramming confirmed the overlapping of these two markers for pluripotency acquisition (Supplementary Fig. 4i). As with *Jmjd3* knockdown, reprogramming efficiency was significantly reduced in *Jmjd3* KO MEFs, and only JMJD3 but not its mutant or UTX overexpression could improve the number of NANOG⁺ colonies (Fig. 2g). This suggested that the two demethylases play distinct roles in reprogramming. The effect of JMJD3 correlated with its activity in H3K27me3 demethylation (Fig. 2h), and was verified using another source of *Jmjd3* KO MEFs[29] (Supplementary Fig. 4j).

Furthermore, we obtained OG2-*Jmjd3*[fl/fl] MEFs and removed the JmjC domain (exons 14–20) with adeno-associated viral Cre[30] (Supplementary Fig. 4k). We used these conditional KO (cKO) MEFs to test the effect of removing *Jmjd3* in early (P2) and late (P5) reprogramming with Vc. As expected, reprogramming efficiency in P2 cKO MEFs was reduced, whereas P5 MEFs reprogrammed at an ~2-fold higher efficiency (Supplementary Fig. 4l). Therefore, the senescence state of the starting MEFs also determines the role of endogenous JMJD3 in reprogramming. In addition, we observed that modulating JMJD3 similarly influences the reprogramming efficiency of neural progenitor cells (NPCs) with Vc (Supplementary Fig. 4m), supporting that JMJD3 function in not restricted to fibroblasts.

Therefore, endogenous JMJD3 is important for the efficient reprogramming of early passage MEFs, and modulating JMJD3 influences the reprogramming of other cell types.

**JMJD3 activates epithelial and pluripotency genes.** We then studied the specific gene networks targeted by JMJD3. To this aim, we performed RNA-sequencing (RNA-seq) at early (day 5) and late (day 10) phases of reprogramming with JMJD3 modulations. We included RNA-seq data of MEFs at P2 with JMJD3 OE or KO and PSCs including iPSCs and ESCs as controls (Fig. 3a). Exogenous JMJD3 showed a remarkable activating effect on gene expression in reprogramming, as the majority of significantly differentially expressed genes (DEGs) at both reprogramming time points were upregulated (80% up at day 5 and 68.3% up at day 10; Supplementary Fig. 5a). The number of genes upregulated by JMJD3 was higher at day 10 than day 5, and the upregulated genes at both time points partially overlapped

(Supplementary Fig. 5b). The tendency with *Jmjd3* knockdown was the opposite (56.3% downregulated genes at day 5 and 61.5% at day 10) to JMJD3 OE, and there was a substantial overlap with genes downregulated by *Jmjd3* knockdown and upregulated by JMJD3 OE (Supplementary Fig. 5a, c).

In order to know how much of the transcriptional effect of JMJD3 modulation in reprogramming comes from its basal effect on MEFs, we correlated the genes upregulated upon JMJD3 OE in OSKM reprogramming at day 5 (OE reprogramming-Up) and the genes reduced by *Jmjd3* knockdown in the same setting (KD reprogramming-Down), with the genes upregulated upon JMJD3 OE (OE-Up) and downregulated by *Jmjd3* cKO (cKO-Down), respectively. We found that ~16% (52 genes) of OE reprogramming-Up and only ~0.5% (4 genes) of KD reprogramming-Down genes overlap with OE-Up and cKO-Down genes in MEFs, respectively (Supplementary Fig. 5d). Among these 52 genes, the majority were MEF-enriched (62%) or transiently activated in reprogramming (25%). These analyses highlighted that the transcriptional impact of JMJD3 modulation is largely context-dependent.

Next, for the 1772 JMJD3-regulated genes in early and late reprogramming, we compared them with MEFs and PSCs, and then divided them into three groups according to the expression pattern: group 1, MEF-enriched (724, ~41%); group 2, transiently activated in reprogramming (603, 34%); and group 3, PSC-enriched (445, 25%) (Fig. 3a). Group 1 and group 2 genes were normally silenced in either early (for group 1) or late (for group 2) reprogramming, and JMJD3 tended to slow down these processes. Gene ontology (GO) analysis for group 2 genes showed categories mainly corresponding to development and differentiation, suggesting a role for JMJD3 in promoting cell plasticity during reprogramming (Supplementary Fig. 5e). Importantly, group 3 genes were related to epithelium and response to LIF, the former gene category pointing to a stronger MET phase in reprogramming[31] (Supplementary Fig. 5f). Individual gene examples included the master epithelial genes *Cdh1*, *Epcam* and *Ocln*, among others (Fig. 3b). Of note, there were also many pluripotency regulators activated by exogenous JMJD3 already at day 5 (e.g., *Sall4*, *Tdh*, and *Sall1*; designated as early-activated), whereas others were activated only at day 10 (e.g., *Dppa5a*, *Utf1*, *Esrrb*, *Mycn*, and *Nanog*; designated as late-activated; Fig. 3c). Similarly, genes downregulated by *Jmjd3* shRNA included epithelial and pluripotency genes (Fig. 3b, c). RT-qPCR confirmed the potent changes of epithelial and pluripotency genes, but not mesenchymal genes, in OSKM reprogramming upon JMJD3 modulation (Supplementary Fig. 5g, h). Western blotting verified the increase of E-cadherin (*Cdh1*) with exogenous JMJD3 (Supplementary Fig. 5i). Moreover,

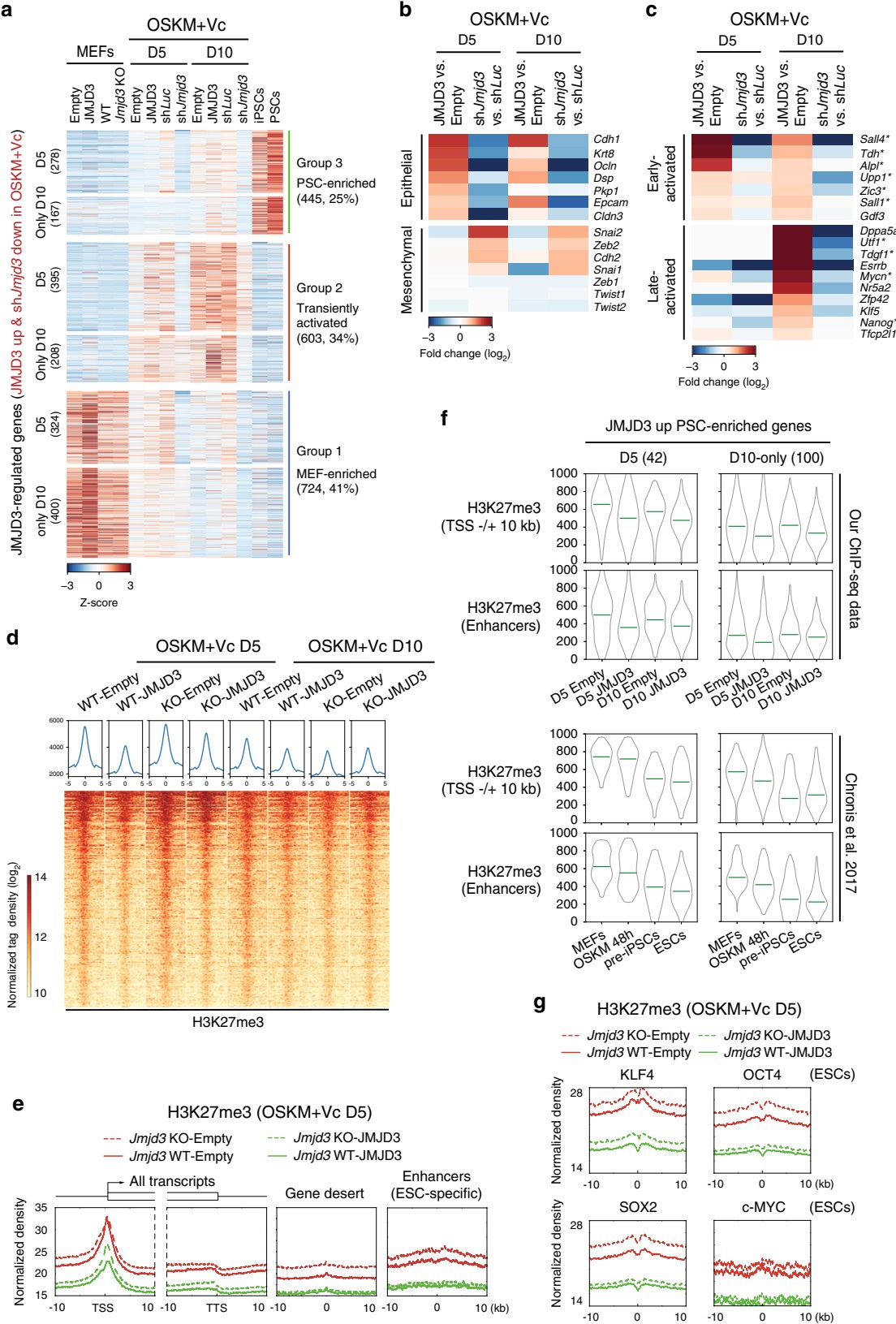

overexpressing JMJD3 but not JMJD3 mutant in *Jmjd3* KO MEFs reprogrammed with OSKM reproduced these findings (Supplementary Fig. 5j).

These results provided a mechanistic explanation for the enhancing effect of JMJD3 in the reprogramming of early passage fibroblasts.

**H3K27me3 demethylation by JMJD3 precedes gene activation.** To study in detail the correlation between changes in gene expression and the reduction in H3K27me3 deposition upon JMJD3 OE in OSKM reprogramming, we performed chromatin immunoprecipitation followed by sequencing (ChIP-seq) with WT and *Jmjd3* KO P2 MEFs at days 5 and 10. Because exogenous

**Fig. 3 JMJD3-induced H3K27me3 demethylation precedes transcription activation. a** RNA-seq heatmap of genes significantly (*q* value < 0.1, fold-change >1.5) upregulated upon JMJD3 OE or downregulated by sh*Jmjd3* #2 at days 5 and 10. These genes were divided into PSC-enriched, transiently activated, and MEF-enriched according to expression pattern in MEFs, PSCs, and OSKM-reprogramming. Left controls: P2 MEFs transduced with Empty or JMJD3, P2 WT, and *Jmjd3* cKO MEFs. Right controls: our own dataset for OSK-iPSCs and PSCs including iPSCs and ESCs from a published dataset (GSE93027). **b** Heatmaps showing the fold change of selected epithelial and mesenchymal genes upon JMJD3 OE or knockdown compared to the controls. **c** Heatmaps showing the fold change of selected early-activated and late-activated pluripotency genes upon JMJD3 OE or knockdown compared to the controls. Genes with asterisk have H3K27me3 demethylation at day 5 of reprogramming. **d** H3K27me3 global density pileups (upper panels) and tag density heatmaps (lower panels) of all H3K27me3 peaks in WT and *Jmjd3* KO P2 MEFs transduced with OSKM and Empty or JMJD3 in medium with Vc at days 5 and 10. **e** Tag density pileups of H3K27me3 in WT (solid line) and *Jmjd3* KO (dashed line) P2 MEFs transduced with OSKM and JMJD3 (green) or Empty (red) at day 5. -/+10 kb windows of regions around the TSS and TTS (left panel), gene desert (middle panel), and within ESC-specific enhancers[34] (including typical enhancers and super-enhancers; right panel), are shown. **f** Violin plots of H3K27me3 levels at TSS and enhancers of PSC-enriched genes that upregulated by JMJD3 OE at day 5 (upper left panels) or only at day 10 in the indicated samples (upper right panels). Numbers in the brackets are gene number for each time point. The lower two panels are reanalysis of the dataset by Chronis et al.[34] (GSE90895). 48 h, 48 hours. **g** Tag density pileups of H3K27me3 in WT and *Jmjd3* KO P2 MEFs transduced with OSKM and Empty or JMJD3 in medium with Vc at day 5 at binding sites of OSKM in ESCs (from GSE11431).

JMJD3 decreases the global levels of H3K27me3 (see above Fig. 2a), we included a spike-in of *Drosophila* genomic DNA for normalization[32] (Supplementary Fig. 6a). As anticipated, exogenous JMJD3 reduced the intensity of the H3K27me3 ChIP-seq signal at day 5 in both WT and *Jmjd3* KO cells (Fig. 3d). Demethylation in *Jmjd3* KO cells at day 5 was slower than in WT cells, suggesting that endogenous JMJD3 also participates in this process. Demethylation upon JMJD3 OE was less prominent or even disappeared at day 10, because by then the global levels of H3K27me3 had already dropped significantly even with empty vector (Fig. 3d). Therefore, we mainly focused on day 5 for further analyses. We also observed that WT and *Jmjd3* KO cells overexpressing JMJD3 shared a large proportion of demethylated loci, though the effect of JMJD3 in WT cells and on day 5 was more widespread (Supplementary Fig. 6b). Genome-wide analysis showed that H3K27me3 peaks at day 5 were more abundantly (>60%) located within a ±5 kb window of the transcription start site (TSS), and exogenous JMJD3 did not change the peak distribution compared to the empty vector (Supplementary Fig. 6c). Exogenous JMJD3 reduced the average genome-wide peak intensity around the TSS, the transcription termination site (TTS) and gene desert regions (Fig. 3e). This pattern was comparable between WT and *Jmjd3* KO cells, though the latter cells displayed higher signal intensity than WT cells (Fig. 3e). Notably, we detected substantial enrichment of H3K27me3 at ESC-specific enhancers[33], which was also potently reduced by exogenous JMJD3 (Fig. 3e). In summary, changes in H3K27me3 induced by JMJD3 in reprogramming happen globally at multiple genome regions.

We tested the correlation between the changes in H3K27me3 at TSS and enhancer regions and the expression levels measured by RNA-seq of JMJD3-upregulated PSC-enriched genes. As expected, genes upregulated by JMJD3 at day 5 showed a clear reduction of H3K27me3 levels at day 5 (Fig. 3f, upper left panels). Using the ChIP-seq dataset by Chronis et al.[34], we observed that these genes display high H3K27me3 in MEFs and an H3K27me3-to-H3K27ac switch in 48 h OSKM reprogramming (Fig. 3f, lower left panels and Supplementary Fig. 6d, upper panels). Analysis of individual H3K27me3 tracks (*Ink4a*/*Arf* is shown as control) and ChIP-qPCR of OSKM reprogramming samples for epithelial loci (*Cdh1* and *Epcam*) and early-activated pluripotency loci (*Tdh*, *Sall1*, and *Sall4*) confirmed this (Supplementary Fig. 6e–g). Interestingly, PSC-enriched genes upregulated by JMJD3 only at day 10 showed H3K27me3 demethylation at day 5 (Fig. 3f, upper right panels). These genes also experienced an H3K27me3-to-H3K27ac switch in the conversion between MEFs and iPSCs, albeit to a lesser extent than at day 5 (Fig. 3f, lower right panels and Supplementary Fig. 6d, upper panels). Individual H3K27me3

tracks for a panel of late-activated pluripotency genes (*Mycn*, *Utf1*, *Nanog*, *Tfcp2l1*, and *Tdgf1*) supported these observations, while for some others (*Dppa5a*, *Dppa4*, *Zfp42*, *Lin28a*, and *Nr5a2*) did not (Supplementary Fig. 7a). These results demonstrate that JMJD3-induced H3K27me3 demethylation at target loci correlates with their activation, though not all gene activation is directly caused by H3K27me3 demethylation.

To see whether changes of other repressive epigenetic modifications may also be involved in JMJD3-mediated pluripotency gene activation, especially late-activated genes, we looked at published H3K9me3[34] and DNA methylation data[35–37]. H3K9me3 levels were relatively low at these loci in MEFs and did not change much in the reprogramming of MEFs to iPSCs (Supplementary Fig. 6d, middle panels). By contrast, DNA methylation at enhancers of both early-activated and late-activated PSC-enriched genes was strikingly high in MEFs and low in ESCs/iPSCs, suggesting that H3K27me3 and DNA demethylation are coordinated during reprogramming (Supplementary Fig. 6d, lower panels and Supplementary Fig. 7a).

In addition, we performed de novo motif discovery within day 5 JMJD3-dependent H3K27me3 demethylated regions in WT cells and discovered motifs for multiple pluripotency factors, including OSKM themselves, STAT3, ESRRB, NR5A2, and PRDM14 (Supplementary Fig. 7b). Comparison of these demethylated regions with available ChIP-seq data for pluripotency transcription factors in ESCs[38–40] showed good correlation, in particular with KLF4 (Supplementary Fig. 7c). Similarly, compared with the other three reprogramming factors, KLF4-bound sites in ESCs displayed the highest levels of H3K27me3 in both WT and *Jmjd3* KO OSKM reprogramming cells (Fig. 3g). This led us to speculate that JMJD3 cooperates with KLF4 to remove H3K27me3 from epithelial and pluripotency loci during reprogramming.

**KLF4 and JMJD3 cooperate to activate target genes**. To study how JMJD3 accesses target loci, we co-immunoprecipitated FLAG-tagged JMJD3 and the OSKM factors at day 5 of P2 MEF reprogramming. As anticipated, we found a strong interaction of JMJD3 only with KLF4 (Fig. 4a). We tested whether exogenous JMJD3 enhances the reprogramming of P2 MEFs transduced with OCT4, SOX2, c-MYC (OSM), and low levels of KLF4, as a basal level of KLF4 is strictly required to obtain GFP$^+$ colonies and avoid cell death. JMJD3 only enhanced reprogramming in the presence of high levels of KLF4 (Fig. 4b), supporting that both proteins work in tandem. Consistently, JMJD3 reduced H3K27me3 at epithelial (*Cdh1* and *Epcam*) and pluripotency loci (*Sall1*, *Sall4*, and *Tdh*), and enhanced their

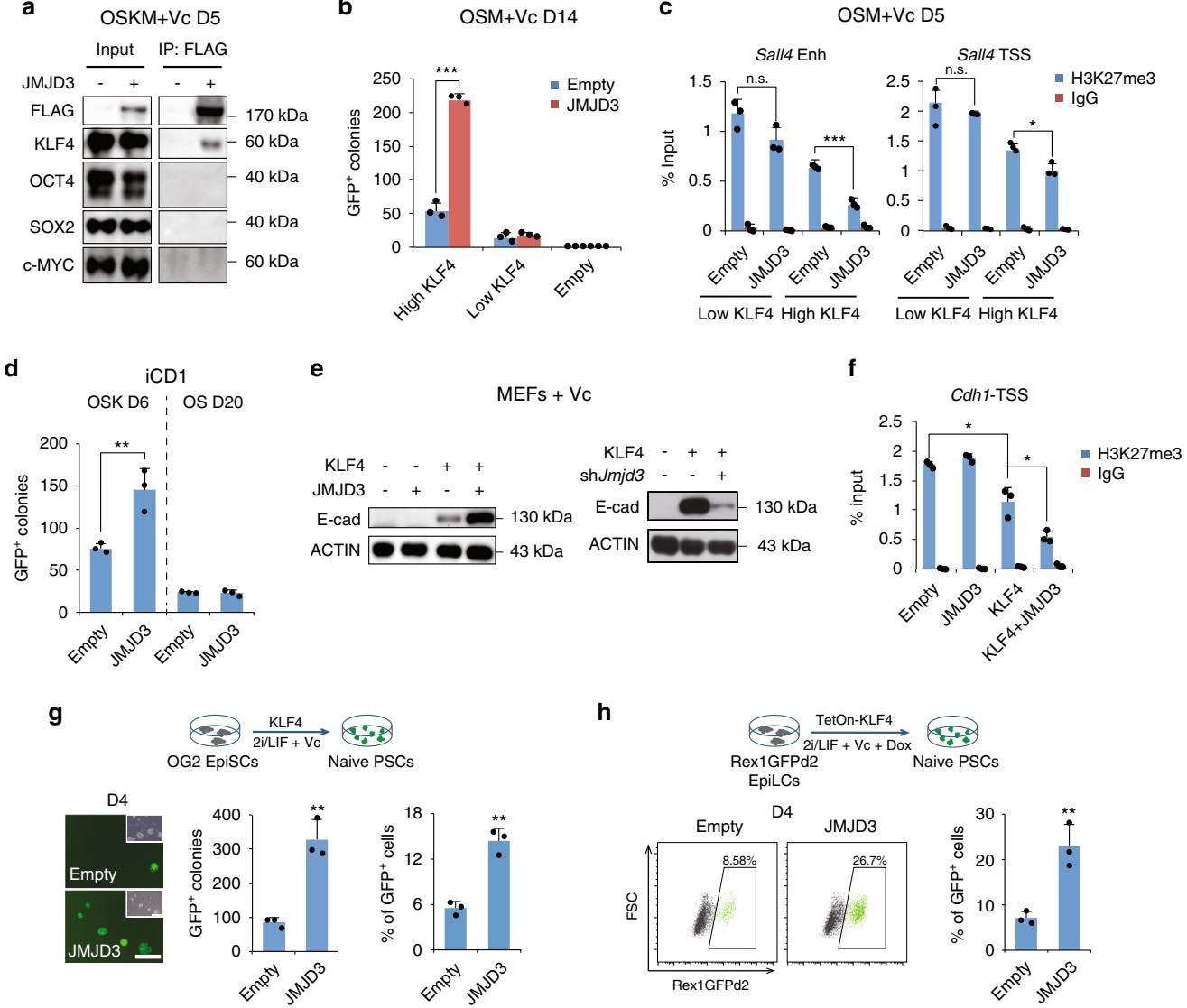

**Fig. 4 JMJD3 cooperates with KLF4 to activate target genes. a** Immunoprecipitation with anti-FLAG beads in nuclear extracts from OSKM-reprogrammed P2 MEFs with FLAG-tagged JMJD3 or Empty at day 5, followed by western blotting with the indicated antibodies. **b** Number of GFP+ colonies at day 14 in P2 MEFs reprogrammed with OSM (OCT4, SOX2, and c-MYC) and the indicated levels of KLF4 (high and low levels represent normal and 1/5 volume of retrovirus, respectively) plus Empty or JMJD3. **c** ChIP-qPCR for H3K27me3 at TSS and enhancer (Enh) of *Sall4* in P2 MEFs reprogrammed with OSM and low or high level of KLF4 plus Empty or JMJD3 at day 5. **d** Number of GFP+ colonies in P2 MEFs reprogrammed with OSK (OCT4, SOX2, and KLF4) at day 6 or OS (OCT4 and SOX2) at day 20 plus Empty or JMJD3 in iCD1 medium. **e** Western blotting for E-cadherin (E-cad) in P2 MEFs transduced with the indicated constructs at day 5. **f** ChIP-qPCR for H3K27me3 at TSS of *Cdh1* in P2 MEFs transduced with the indicated constructs in medium with Vc at day 5. **g** Schematic depiction of the EpiSC-to-naïve PSC (Epi-to-naïve) transition using KLF4 in OG2 EpiSCs (upper panel); images and number of GFP+ colonies and also percentage of GFP+ cells measured by flow cytometry at day 4 with Empty or JMJD3 (lower panel). Scale bar, 250 μm. **h** Schematic depiction of the Epi-to-naïve transition in Rex1GFPd2 epiblast-like cells (EpiLCs) using KLF4 (upper panel), the percentage of GFP+ cells measured by flow cytometry at day 4 with Empty or JMJD3 (lower panel). Error bars represent the s.e.m. Data are presented as mean ± s.e.m. from *n* = 3 **b–d**, **f–h** biologically independent experiments. Statistical analyses were performed using a two-tailed unpaired Student's *t*-test (*P < 0.05; **P < 0.01; ***P < 0.001). *P* values: 2.72 × 10⁻⁵ **b**; 0.0002, 0.0432 **c**; 0.009 **d**; 0.0136, 0.0169 **f**; 0.0021, 0.0012 **g**; 0.0056 **h**. Experiments were repeated independently three times **a** or twice **e** with similar results. Source data are provided as a Source Data file.

transcriptional activation and protein expression, only in reprogramming with high levels of KLF4 (Fig. 4c and Supplementary Fig. 8a–c). JMJD3 also enhanced P2 MEF reprogramming with OCT4, SOX2, and KLF4 (OSK) with or without Vc, albeit moderately (Supplementary Fig. 8d). Furthermore, we used iCD1 medium, which can induce reprogramming with OCT4 and SOX2 alone[41]. JMJD3 enhanced GFP+ colony formation with OSK in iCD1 medium, but could not synergize with OCT4 and SOX2 (Fig. 4d).

KLF4 overexpression alone induces an epithelial gene program in MEFs[18]. To test whether JMJD3 cooperates with KLF4 to achieve this effect, we overexpressed either KLF4, JMJD3, or both in P2 MEFs. JMJD3 magnified the enhancing effect of KLF4 on *Cdh1*, *Epcam*, and *Ocln* expression, as measured by RT-qPCR and western blotting (for E-cadherin), whereas alone had no effect (Fig. 4e, left panel and Supplementary Fig. 8e). Similarly, *Jmjd3* knockdown reduced the enhancement of epithelial genes by KLF4 (Fig. 4e, right panel and Supplementary Fig. 8f). Moreover, ChIP-qPCR analysis

of the *Cdh1* locus showed that exogenous KLF4 alone moderately reduces H3K27me3 levels, but combined with exogenous JMJD3 the effect is stronger (Fig. 4f). Consistent with previous reports[22], JMJD3 but not KLF4 could induce *Ink4a/Arf* expression and H3K27me3 demethylation of the locus, and their combination was not synergistic (Supplementary Fig. 8e, g).

KLF4 is more highly expressed in naïve (pre-implantation like) mouse ESCs than primed pluripotent (or post-implantation epiblast-like) stem cells (EpiSCs), and can induce the transition of the latter into the former cell type[19]. Thus, we studied whether JMJD3 plays a role in promoting this transition. Exogenous JMJD3 substantially enhanced the transition, as judged by both GFP+ colony quantification and flow cytometry, with upregulation of naïve-specific genes (Fig. 4g and Supplementary Fig. 8h). We could further confirm this effect with Rex1GFPd2 reporter cells[42] (Fig. 4h).

Altogether, these findings highlight a role for the KLF4–JMJD3 axis in multiple cell fate transitions.

**KLF4 and JMJD3 cooperate genome wide during reprogramming**. To gain a genome-wide view of the function of the KLF4–JMJD3 tandem, we performed ChIP-seq for exogenous JMJD3 in OSM reprogramming cells in the presence of high or low levels of KLF4. JMJD3 binding centered mostly around the TSS but also included distal regions (Supplementary Fig. 9a). As shown in Fig. 5a, 86.5% (11,860) of JMJD3 peaks with high KLF4 were lost in the low KLF4 condition. De novo motif discovery in the JMJD3 peaks identified OSK motifs (Fig. 5b), which is expected considering that the three reprogramming factors often bind to chromatin in combination[43]. Moreover, the percentage of sites with a KLF4 motif in the low-KLF4 condition was substantially reduced (Fig. 5b). We verified the ChIP-seq results with ChIP-qPCR (Supplementary Fig. 9b).

Next, we did ChIP-seq for KLF4 at day 5 of OSKM reprogramming with empty vector or JMJD3. This yielded 13,966 KLF4-binding sites, >70% of which contained motifs resembling the canonical KLF4 motif (Fig. 5c and Supplementary Fig. 9c). Exogenous JMJD3 did not substantially alter the genomic distribution of KLF4, as ~95% of the KLF4 peaks were shared between empty vector and JMJD3 overexpressing cells (Fig. 5c). H3K27me3 levels were reduced in the regions corresponding to KLF4 peaks when JMJD3 was overexpressed (Fig. 5d, left panel). Moreover, although globally there was only a modest increase in KLF4 binding to chromatin with exogenous JMJD3, epithelial and pluripotency loci including *Cdh1*, *Sall4*, *Sall1*, and *Tdh* showed a significant increase, which we verified with ChIP-qPCR (Fig. 5d, right panel and Supplementary Fig. 9d, e).

To understand more clearly the relationship between KLF4 and JMJD3, we compared their binding sites. As shown in Fig. 5e, ~39% of JMJD3 and KLF4 peaks overlapped, which we designated as cluster 1. This cluster contained mostly KLF family member motifs and included many epithelial and pluripotency genes. JMJD3-only peaks (cluster 2) showed SOX, OCT, and AP-1 family member motifs, but were less enriched for KLF motifs, suggesting that JMJD3 is also recruited to these loci by other (somatic) transcription factors. KLF4-only peaks (cluster 3) showed almost exclusively KLF family member motifs (Fig. 5f). Interestingly, GO analysis showed that cluster 2 sites include many neural development-related genes as well as other development-related signaling pathways such as WNT and JNK, in striking contrast to cluster 1 or cluster 3 (Supplementary Fig. 9f). Moreover, cluster 2 peaks only opened transiently during reprogramming (Supplementary Fig. 9g). This is consistent with our findings above that JMJD3 OE induces transient activation of developmental genes (Fig. 3a), further reinforcing the idea that

JMJD3 might endorse multi-lineage cell plasticity in reprogramming, as it does in ESC differentiation[12].

Finally, to see how these binding patterns of KLF4–JMJD3 contribute to gene activation, we overlapped early and late-activated DEGs with the genes in clusters 1, 2, and 3. First, KLF4 and/or JMJD3 (all three clusters) bound around half of both early-acivated and late-activated DEGs at day 5 and the binding was slightly more evident for PSC-enriched DEGs (Supplementary Fig. 9h). This confirmed that a significant part of transcriptional changes in reprogramming can be attributed to the direct binding of KLF4 and/or JMJD3. Second, KLF4 and JMJD3 (cluster 1) co-bound more early-activated than late-activated DEGs for both total and PSC-enriched DEGs. Third, KLF4 alone (cluster 3) bound more genes at both stages than JMJD3 alone (cluster 2), and in the late stage the difference was more striking. Noticeably, KLF4 alone could bind 36% of late-activated PSC-enriched genes in early reprogramming. These genes may be activated either through JMJD3 binding in the late stage or an unrelated mechanism.

Our findings not only demonstrate the cooperative binding of KLF4 and JMJD3 at epithelial and pluripotency loci to enhance reprogramming but also show binding to developmental loci. The latter is consistent with the appearance of subpopulations with lineage differentiation potential in reprogramming[24,44].

**JMJD3 promotes enhancer–promoter looping and elongation**. We noticed that although H3K27me3 demethylation happens mainly around the TSS of JMJD3 target genes (Supplementary Fig. 6f, g), JMJD3 and KLF4 bind mostly to their enhancer regions (Supplementary Fig. 9b, d, e). This suggested an enhancer–promoter interaction (looping) mechanism that promotes H3K27me3 demethylation and potentially influences other aspects of transcriptional regulation at JMJD3 target loci. To study this, we performed immunoprecipitation of exogenous JMJD3 followed by mass spectrometry (MS) in OSKM reprogramming of P2 MEFs. We identified substantial enrichment in the cohesin-loading factor NIPBL, a master regulator of enhancer–promoter looping[45], along with other cohesin complex components, the H3K27me2/1 demethylase KDM7A (or KIAA1718), and the enhancer-related remodeler CHD7 (Fig. 6a; the full protein dataset is shown in Supplementary Data 1). We confirmed the interaction of JMJD3 with components of the cohesin complex (NIPBL and SMC1A), and also identified components of another complex essential for enhancer–promoter looping in transcriptional regulation, the mediator complex (MED1 and 12) (Fig. 6b). In contrast, MS analysis of UTX-interacting proteins showed almost exclusive association with the MLL3/4 complex[46] (Fig. 6a, b).

NIPBL is necessary for pluripotency maintenance of mouse ESCs[45], whereas SMC1A is necessary for reprogramming[47]. To test the role of NIPBL in reprogramming and understand how this relates to JMJD3, we immunoprecipitated endogenous NIPBL and confirmed its association with JMJD3 and KLF4 at day 5 (Fig. 6c). ChIP-qPCR of NIPBL verified that JMJD3 enhances its binding to the same loci as KLF4 and JMJD3 (Fig. 6d). Likewise, *Nipbl* knockdown significantly blocked the enhancement of OSKM reprogramming and EpiSC-to-naïve ESC conversion efficiency triggered by exogenous JMJD3 (Fig. 6e, f and Supplementary Fig. 10a), suggesting that an enhancer–promoter looping mechanism is necessary in these contexts.

Immunoprecipitation and western blotting also showed interaction of JMJD3 with the histone acetyltransferase p300, the catalytic subunit of the p-TEFb complex CDK9, and the acetylated histone-reader and CDK9-activator BRD4 (Fig. 6b), all of which occupy enhancers and super-enhancers[33]. Using

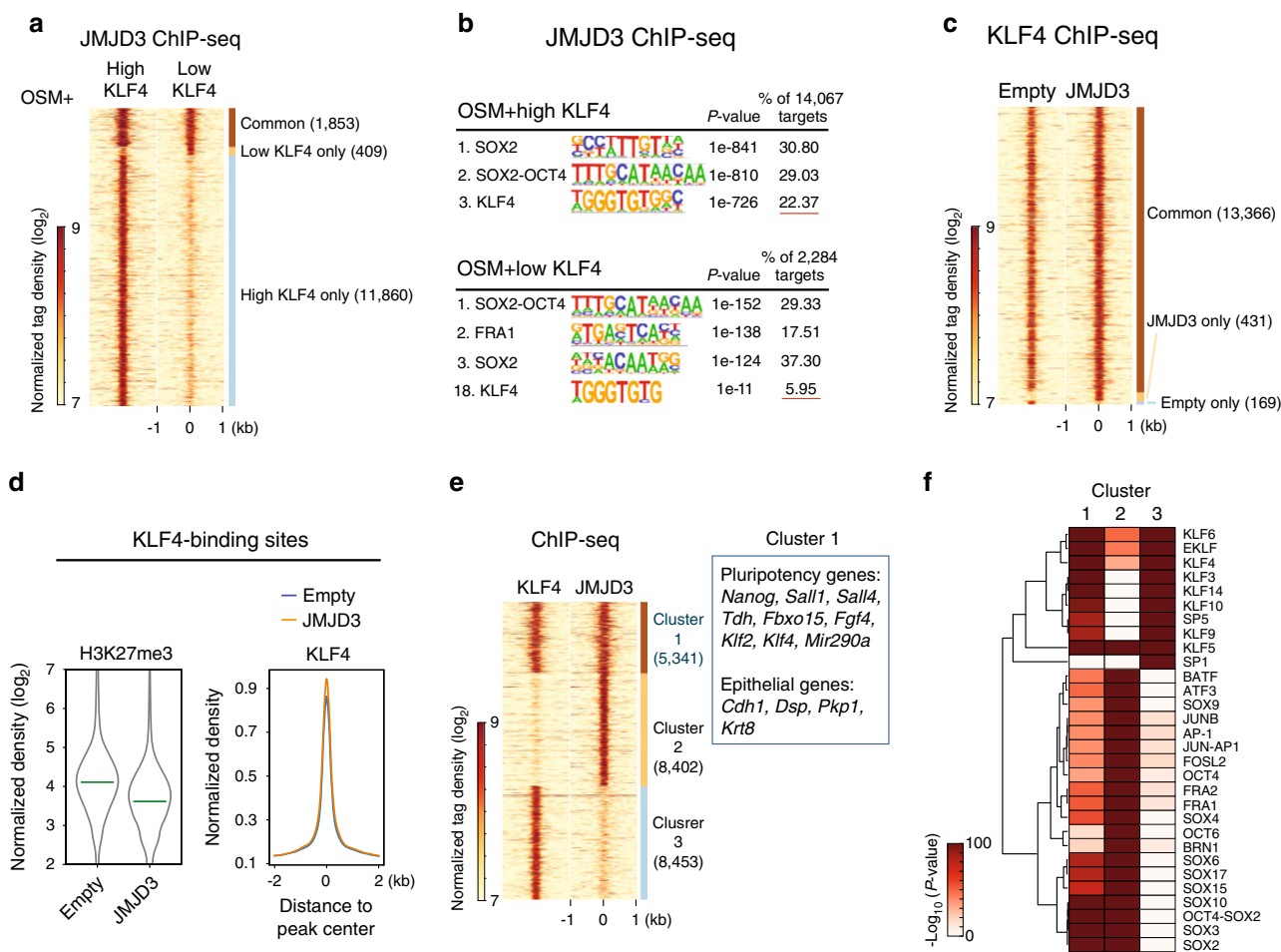

**Fig. 5 The genome-wide cooperation of KLF4 and JMJD3 during reprogramming. a** Tag density heatmaps of JMJD3 peaks in P2 MEFs transduced with OSM + JMJD3 and low or high level of KLF4 in medium with Vc at day 5. Peaks were merged into the same genomic loci if the summits were within 400 bp, and the resulting merged peaks were allocated to clusters as indicated: common between high and low (Common), low KLF4-specific (Low KLF4 only), and High KLF4-specific (High KLF4 only). **b** De novo motif discovery by HOMER at JMJD3 binding sites. **c** Tag density heatmaps of KLF4 binding in P2 MEFs transduced with OSKM and Empty or JMJD3 in medium with Vc at day 5. Peaks were merged into the same genomic loci if the summits were within 400 bp and the resulting merged peaks were allocated to clusters as indicated: common between Empty and JMJD3 (Common), JMJD3-specific (JMJD3 only), and Empty-specific (Empty only). **d** Violin plots of the levels of H3K27me3 at KLF4-binding sites (left panel) and the pileup tag density plots of KLF4-binding intensity (right panel) with Empty or JMJD3. **e** Tag density heatmaps of JMJD3 or KLF4 peaks. Peaks were merged into the same genomic loci if the summits were within 400 bp, and the resulting merged peaks were allocated to the indicated clusters. Cluster 1: common between JMJD3 and KLF4; Cluster 2: JMJD3-specific; Cluster 3: KLF4-specific. Representative pluripotency and epithelial genes co-bound by KLF4 and JMJD3 in Cluster 1 are listed. **f** Enrichment of specific transcription factor motifs (from HOMER) in each cluster. Data were grouped based on a Euclidean distance matrix and complete linkage.

sucrose gradient ultracentrifugation of nuclear lysates, we confirmed that JMJD3 co-exists with KLF4 and all the above-mentioned co-regulators in fractions 5 and 7 (Supplementary Fig. 10b). Then, we compared the changes in H3K27me3 levels induced by exogenous JMJD3 in reprogramming with ChIP-seq data for cohesin (NIPBL, SMC1A, and SMC3) and mediator components (MED1 and MED12), p300, BRD4, and CDK9 in ESCs[33,45,48,49]. All these co-regulators preferentially bound to sites that lost H3K27me3 in reprogramming upon JMJD3 OE (Supplementary Fig. 10c). Taking the *Sall4* locus as an example, we observed that they bind to H3K27me3 depleted sites at the TSS and enhancer regions of ESCs but not of MEFs (Supplementary Fig. 10d). ChIP-qPCR verified that exogenous JMJD3 enhanced the loading of cohesin and mediator components, and CDK9, on enhancers and promoters of epithelial and pluripotency regulators at day 5 of reprogramming (Supplementary Fig. 10e). Importantly, knockdown experiments showed that

endogenous JMJD3 is needed for efficient binding of KLF4 and cohesin loading (Supplementary Fig. 10f).

To test the functional consequence of the interaction between JMJD3 and these additional co-regulators, especially p300, we performed ChIP-seq for H3K27ac. We observed an increase of H3K27ac around the TSS of upregulated PSC-enriched genes upon JMJD3 OE at day 5 (Fig. 6g). This is consistent with our above analysis of published datasets indicating an H3K27me3-to-H3K27ac switch at these loci during normal reprogramming (see Supplementary Fig. 6d). Changes induced by JMJD3 OE in H3K27ac at KLF4-bound sites at day 5 reprogramming anti-correlated with the changes in H3K27me3 (Fig. 6h), as shown at representative epithelial (*Cdh1* and *Epcam*) and pluripotency (*Sall1*, *Sall4*, and *Tdh*) loci by ChIP-seq and ChIP-qPCR (Supplementary Fig. 11a, b). Conversely, JMJD3 depletion significantly reduced H3K27ac at these loci (Supplementary Fig. 11c). The H3K27me3-to-H3K27ac switch on both promoters

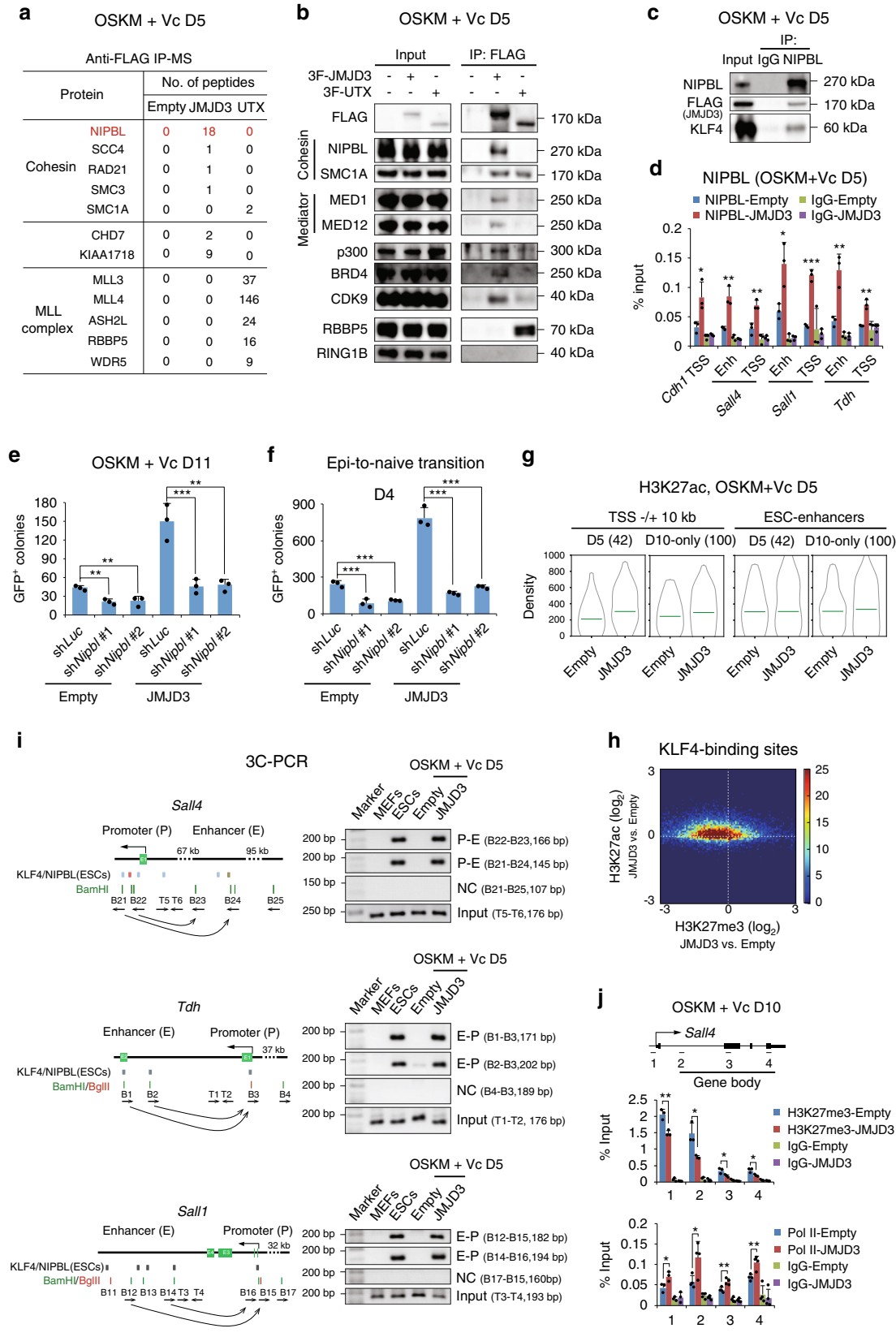

and enhancers implied enhancer activation and enhancer-promoter looping[50]. We used chromosome conformation capture (3C) technology to determine whether JMJD3 facilitates enhancer-promoter looping in reprogramming. For all three selected early-activated pluripotency genes (Sall4, Tdh and Sall1), ESC-specific looping could only be detected in OSKM with JMJD3 but not in OSKM control or in MEFs (Fig. 6i and Supplementary Fig. 11d). In addition, ChIP-qPCR showed increased RNA polymerase II (Pol II) binding by JMJD3 OE at the TSS and gene body of the same epithelial and pluripotency target loci, indicating increased Pol II recruitment and productive transcriptional elongation (Fig. 6j and Supplementary Fig. 11e).

**Fig. 6 JMJD3 facilitates enhancer–promoter looping and transcriptional activation. a**, **b** List of proteins identified by MS **a** and western blotting with indicated antibodies **b** following immunoprecipitation with anti-FLAG beads in nuclear extracts from OSKM-reprogrammed P2 MEFs with FLAG-tagged JMJD3 or UTX. RING1B is the negative control. **c** Immunoprecipitation with NIPBL antibody in nuclear extracts from OSKM-reprogrammed P2 MEFs with FLAG-tagged JMJD3, followed by western blotting with the indicated antibodies. **d** ChIP-qPCR for NIPBL at the enhancer/TSS of the indicated genes in OSKM-reprogrammed P2 MEFs with Empty or JMJD3. **e** Number of GFP+ colonies in OSKM-reprogrammed P2 MEFs with Empty or JMJD3 plus sh*Luc* or sh*Nipbl*. **f** Number of GFP+ colonies in OG2 EpiSCs transduced with KLF4 and Empty or JMJD3 plus sh*Luc* or sh*Nipbl*. **g** Violin plots of H3K27ac levels in OSKM-reprogrammed P2 MEFs with Empty or JMJD3 at TSS/enhancers of PSC-enriched genes upregulated upon JMJD3 OE at day 5 or only at day 10. **h** Correlation of H3K27me3 and H3K27ac changes at KLF4-binding sites by JMJD3 OE in OSKM-reprogrammed P2 MEFs with Empty or JMJD3 in medium with Vc at day 5, shown as a 2D density histogram where each point represents a KLF4-binding site. **i** Schematic (left panels) and 3C-PCR (right panels) of enhancer–promoter looping between the indicated sites. Arrows under restriction enzyme sites mark the orientation of the primers. **j** ChIP-qPCR for H3K27me3 and Pol II at the indicated loci of *Sall4* in OSKM-reprogrammed P2 MEFs with Empty or JMJD3. Error bars represent the s.e.m. Data are presented as mean ± s.e.m. from n = 3 **d**–**f**, **j** biologically independent experiments. Statistical analyses were performed using a two-tailed unpaired Student's *t*-test **d**, **j** or one-way ANOVA followed by a Dunnett multiple comparison test **e**, **f** (*$P < 0.05$; **$P < 0.01$; ***$P < 0.001$). *P* values: 0.0343, 0.0059, 0.0059, 0.0228, 0.0001, 0.0062, 0.0019 **d**; 0.0049, 0.0053, 0.0009, 0.0010 **e**; 0.0004, 0.0007, 0.0001, 0.0001 **f**; 0.0054, 0.0193, 0.0364, 0.0161 (H3K27me3), 0.0419, 0.0288, 0.0074, 0.0063 (Pol II) **j**. Experiments were repeated independently three times **b** or twice **c**, **j** with similar results. Source data are provided as a Source Data file.

Therefore, JMJD3 and KLF4 form complexes with other co-regulators to control multiple aspects of transcriptional regulation in reprogramming, in particular enhancer-promoter looping, and all these functions are coordinated with the removal of H3K27me3.

## Discussion

We have shown that JMJD3 impacts somatic cell reprogramming in two opposing ways. The detrimental effects of inducing the pro-senescence regulator *Ink4a*/*Arf* and degrading the pluripotency factor PHF20 are OSKM independent. In contrast, JMJD3-mediated activation of epithelial and pluripotency genes is KLF4 dependent. This consideration is important because it helps to clarify that the reprogramming-specific function of JMJD3, like that of UTX, is to boost the process. This enhancement can be facilitated by using early passage donor cells or potentiating JMJD3 catalytic activity with Vc, but blocked when using late passage donor cells (Fig. 7a). Despite their functional similarity, JMJD3 loss-of-function cannot be rescued by UTX, nor vice-versa[14]. Both H3K27me3 demethylases may either be recruited to different loci or interact with different sets of co-regulators. In this regard, UTX mainly interacts with MLL but not NIPBL or mediator[46].

KLF4 recruits JMJD3 to target loci and in turn JMJD3-mediated H3K27me3 demethylation cooperatively facilitates the binding of KLF4 to chromatin. DNA demethylation also seems to be involved in the activation of PSC-enriched genes upregulated by JMJD3. In fact, it was recently reported that KLF4 also interacts with the DNA demethylase TET2 to promote reprogramming[51]. This suggests that the coordinated effects of inducing H3K27me3 demethylation and DNA demethylation underlie the pioneer factor role of KLF4 in reprogramming. Besides removing H3K27me3, JMJD3 participates in further critical steps of transcriptional regulation such as the reorganization of 3D chromatin interactions and productive transcriptional elongation (Fig. 7b). The former is driven by the dissolution of somatic H3K27me3-dependent and PRC2-mediated long chromatin interactions[52], and by the recruitment of NIPBL that allows enhancer–promoter looping. The substitution of H3K27me3 for H3K27ac through recruitment of p300 likely contributes to sustaining enhancer–promoter looping and to the increased transcriptional flux by facilitating co-activator complex formation[53]. Notably, KLF4 has been previously linked to all these aspects of transcriptional regulation in reprogramming[54–56], supporting our model that JMJD3 and KLF4 act in tandem, with JMJD3 facilitating the pioneer factor role of KLF4. However, JMJD3 is not indispensable for reprogramming, as iPSCs could still be generated from *Jmjd3* KO MEFs albeit with lower efficiency. Other factors likely complement the function of JMJD3 in the above-mentioned mechanisms during reprogramming.

KLF4, BRD4, and CDK9 were recently found to be upregulated in B cells transiently exposed to C/EBPα, and mediate elite cell reprogramming[57]. Similarly, we found that KLF4 and JMJD3 regulate reprogramming of NPCs. Hence, it is plausible that the KLF4–JMJD3 tandem regulates the reprogramming of other cell sources. In addition to reprogramming, we have shown that KLF4 and JMJD3 cooperate to induce an MET program in fibroblasts. Similarly, JMJD3 facilitates the KLF4-driven EpiSC-to-naïve PSC transition, a context in which chromatin interactions are rearranged[58], through NIPBL. We also observed that JMJD3-only peaks (cluster 2) are enriched in genes of development function and transiently open, suggesting that JMJD3 might endorse multi-lineage cell plasticity in reprogramming.

Overall, our work provides a new integrative picture for the roles of KLF4 and JMJD3 in multiple cell fate transitions. It remains to be studied whether this phenomenon applies to other in vitro and in vivo cell settings, such as transdifferentiation, epithelial homeostasis, cancer or embryonic development.

## Methods

**Animals, cells and culture conditions**. The use of mice in this study was approved by the Institutional Animal Care and Use Committee of Guangzhou Institutes of Biomedicine and Health, Chinese Academy of Sciences, under license number 2014013. Mice were housed in a specific pathogen-free environment with a 12-hour light/dark cycle. Temperature was maintained at 22–24 °C with a relative humidity of 40–70%. Euthanasia was performed by carbon dioxide inhalation. OG2 MEFs including OG2 *Jmjd3* cKO MEFs were isolated from E13.5 embryos carrying the *Oct4*-GFP transgene[4,59] and used for all experiments unless otherwise indicated. *Jmjd3* cKO mice were purchased from Model Animal Research Center Of Nanjing University[30] and mated with OG2/129 mice. The offspring with *Jmjd3*fl/fl and *Oct4*-GFP transgene was used for MEF isolation. OG2 TTFs were isolated from infant mice ~2 weeks after birth using standard procedure. OG2/*Dppa5a*-tdTomato dual reporter MEFs were isolated from E13.5 embryos of a *Dppa5a*-tdTomato knockin reporter mice in OG2 background[24]. All TTFs and MEFs, including the two sources of *Jmjd3* traditional KO MEFs[28,29], were maintained in DMEM/high glucose (Hyclone) containing 10% fetal bovine serum (FBS, NTC), GlutaMax (Invitrogen), nonessential amino acids (NEAA, Invitrogen), and penicillin/streptomycin (Hyclone). HEK293T and Plat-E cells were maintained in DMEM/high glucose containing 10% FBS. Mouse ESCs, JMJD3 OE-iPSCs and pre-iPSCs (clone # pre 2–2)[60] were cultured on feeder layers (mitomycin-C-treated MEFs) in mouse ESC medium containing DMEM/high glucose (Hyclone) supplemented with 15% FBS (GIBCO), GlutaMax, NEAA, sodium pyruvate, penicillin/streptomycin, β-mercaptoethanol, and LIF (Millipore). OG2 ESCs used in ATAC-seq were cultured on gelatin in 2i/L medium (F12/Neurobasal supplemented with 0.5× N2 (GIBCO), 0.5× B27 (GIBCO), GlutaMax, NEAA, sodium pyruvate, β-mercaptoethanol, LIF, 3 μM CHIR99021, and 1 μM PD0325901), whereas OSK-iPSCs used in RNA-seq and western blotting were maintained on feeder layers in mESC medium

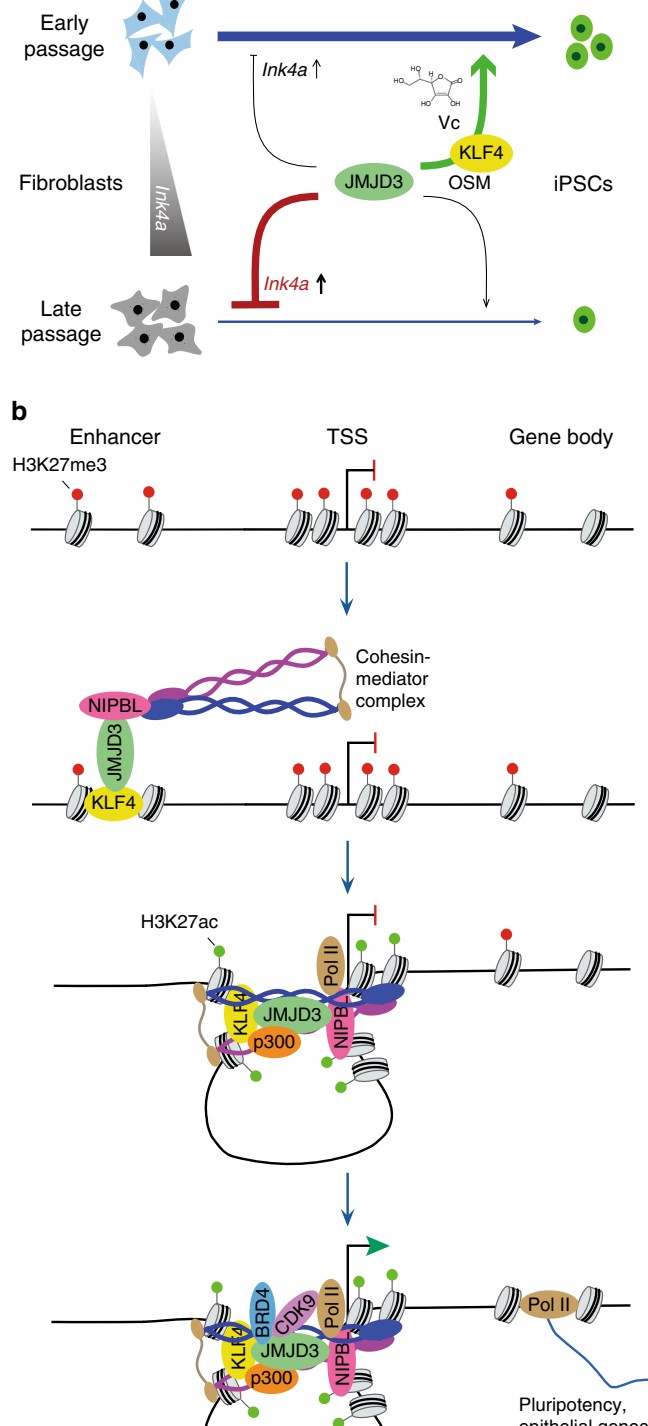

**Fig. 7 Working model for JMJD3 and KLF4 in reprogramming. a** JMJD3 positively and negatively regulates reprogramming in early and late passage fibroblasts respectively. **b** KLF4 recruits JMJD3 and co-factors to induce gene activation in reprogramming.

supplemented with 3 μM CHIR99021 and 1 μM PD0325901. Mouse OG2 EpiSCs were derived from E5.5 embryos generated by mating homozygous OG2 male mice with 129/Sv female mice[61], and initially passaged on feeders before culturing on fibronectin-coated or FBS-coated plates in N2B27 bFGF/Activin A medium containing F12/Neurobasal (1:1, GIBCO) supplemented with 0.5× N2 (GIBCO), 0.5× B27 (GIBCO), GlutaMax, NEAA, BSA, β-mercaptoethanol, 20 ng/ml Activin A

(R&D), and 20 ng/ml bFGF (Peprotech). NPCs were isolated from the brain of E13.5 mouse embryos under a dissection microscope[62]. Briefly, mouse meninges and blood vessels were removed and discarded. The remaining brain tissues were sliced into small pieces and dissociated with 0.15% trypsin (GIBCO) for 15 min at 37 °C and then plated onto T25 flask bottle. NPCs were maintained in NPC medium containing DMEM/F12 (1:1), 1× N2, 1× B27, 20 ng/ml EGF (PeproTech) and 20 ng/ml bFGF. *Drosophila* S2 cells (Invitrogen, R69007) were cultured in Schneider's *Drosophila* medium (Life technologies, 21720-024) supplemented with 10% FBS (NTC). All cells were routinely checked for mycoplasma contamination (MycoAlert, Lonza).

**iPSC generation and EpiSC-to-naïve PSC transition.** OSKM and OSK reprogramming of fibroblasts was performed in mouse ESC medium with or without 50 μg/ml Vc (Sigma, A4034), or where indicated in iSF1 medium[25] or iCD1 medium[41]. The latter two media were prepared as reported. For OS reprogramming, 10 ng/ml BMP4 was additionally included in iCD1 medium. Plat-E cells were transfected with individual OSKM pMXs vectors to produce retroviral supernatants[4]. MEFs and TTFs were plated at 4000–5000 cells per square centimeter (~15,000 cells per well of 12-well plate) and transduced with the retroviruses. We designated the first infection time point as day 0. After two rounds of 24-h infection, the medium was changed to mouse ESC medium and Vc was included in the medium where indicated. Around day 11 (with Vc) or day 16 (without Vc), GFP+ or NANOG+ colonies were counted using a fluorescence microscope. For AP staining, cells were washed twice with PBS and fixed with 4% paraformaldehyde for 2 min. Then after washing with TBST (50 mM Tris–HCl at pH 8.0, 150 mM NaCl and 0.1% Tween 20), cells were incubated with freshly prepared AP staining solution (4.5 μl 50 mg/ml nitro blue tetrazolium, 3.5 μl 50 mg/ml 5-bromo-4-chloro-3-indolyl phosphate in 100 mM Tris–HCl, pH 9.5, 100 mM NaCl, 50 mM MgCl₂) in the dark for 15 min at room temperature and washed twice with PBS. Data were collected with Image Pro Plus 6.0 after scanning. Lentiviruses for shRNAs constructed in pLKO.1 vector were packaged in HEK293T cells and diluted 1:10 with fresh medium for infection. One round of 8-h lentivirus infection was performed after OSKM infection. For the reprogramming of *Jmjd3* cKO MEFs, *Jmjd3*[fl/fl] MEFs at P1 were infected with Cre or control RFP adeno-associated viruses (PackGene) for 24 h to complete JMJD3 JmjC domain deletion and then these cells were reprogrammed to pluripotency colonies in standard way. For NPC reprogramming, P3 NPCs were first infected with OKM retroviruses in NPC medium for 12 h and 2 days later with JMJD3 retroviruses or sh*Jmjd3* lentiviruses. Then, they were plated on feeders in mouse ESC medium with Vc for reprogramming. Pre-iPSC reprogramming was performed in mouse ESC medium with 50 μg/ml Vc. For EpiSC-to-naïve PSC transition, OG2 mouse EpiSCs in suspension were transduced with KLF4 lentiviruses (1/5 dilution) for 8 h with ROCK inhibitor Y-27632 (Selleck) and treated with G418 24 h later. After 3 days of selection, the cells were split upon dissociation with TrypLE, and on the next day, the medium was changed to 2i/LIF medium and 50 μg/ml Vc to induce naïve PSCs. For the EpiSC-to-naïve PSC transition of Rex1GFPd2-EpiLCs, Rex1GFPd2-ESCs were differentiated into EpiLCs in N2B27 bFGF/Activin A medium, stabilized for more than five passages, and then transfected using Lipofectamine 3000 (Invitrogen) with piggyBac-TRE-*Klf4*-EF1α-rtTA plus pBase vector. After three-passage selection with Blasticidin (InvivoGen), Rex1GFPd2-EpiLCs were converted in 2i/LIF with both Vc and Doxycycline (Dox). The efficiency of EpiSC-to-naïve PSC transition was measured by counting the colonies and by flow cytometry.

**Plasmids and molecular cloning.** pMXs retroviral vectors separately expressing OSKM were purchased from Addgene. Full-length mouse *Jmjd3*, *Utx*, and *Phf20* were amplified and cloned into a pMXs backbone vector containing three tandem FLAG (3×FLAG) sequences. JMJD3 mutant and shRNA rescue mutants were cloned after PCR with site-directed mutagenesis primers. For shRNA vectors, oligonucleotides containing the target sequence were annealed and cloned into pLKO.1 vector (Addgene). Target sequences for shRNAs are listed in Supplementary Data 2.

**RNA isolation and RT-qPCR.** Total RNA was extracted from cells using RNAzol® RT (MRC) following the manufacture's instructions. 2 μg of RNA were subjected to reverse transcription. qPCR was performed with an ABI 7500 machine using SYBR green (Takara). Assays were run in triplicate and values were normalized on the basis of *Actb* expression values. At least three independent experiments were performed, and data were collected with real-time PCR software v2.4. RT-qPCR primers are listed in Supplementary Data 2.

**Immunofluorescence.** For immunofluorescence, cells on coverslips or plates were fixed with 4% paraformaldehyde for 30 min, permeabilized with 0.2% Triton X-100 in PBS for 30 min, and blocked with 3% bovine serum albumin for 2 h. Cells were then incubated with primary antibodies overnight at 4 °C, and subsequently with secondary antibodies for 1 h after washing with PBS. Finally, cells were stained with DAPI for 1–2 min and observed with a confocal microscope. Photos were captured with ZEN software v2.0. Primary antibodies were: anti-FLAG (Sigma, F1804), anti-H3K27me3 (Millipore, 07-449), and anti-NANOG (BETHYL A300-397A). The dilution for the primary antibodies was 1/200.

**Western blotting and co-immunoprecipitation.** For western blotting, cells were washed with PBS and lysed on ice in RIPA buffer (50 mM Tris–HCl pH 7.4, 150 mM NaCl, 0.25% sodium deoxycholate, 0.1% NP-40, and 0.1% Triton X-100) supplemented with protease inhibitor cocktail (Roche). Samples were subjected to SDS–PAGE and transferred onto a PVDF membrane (Millipore). Membranes were blocked with 5% nonfat milk in TBST, and then sequentially incubated with primary and secondary antibodies. Signals were detected by Amersham ECL (GE Healthcare), visualized with the FUSION SOLO 4M machine (Vilber Lourmat) and analyzed with FusionCapt Advance Solo4.16.15. For co-immunoprecipitation, nuclear extracts were prepared with NE-PER™ Nuclear and Cytoplasmic Extraction Reagents (Thermo Scientific, 78833). Nuclear lysates (from $2 \times 10^7$ cells) were incubated overnight at 4 °C with anti-FLAG M2 magnetic beads (30 μl; Sigma, M8823) for 3×FLAG-tagged JMJD3/UTX or NIPBL antibodies (10 μg) conjugated to Protein G Dynabeads (100 μl; Invitrogen) for NIPBL endogenous immunoprecipitation. Beads were washed 5–8 times with washing buffer (50 mM Tris–HCl pH 7.9, 100 mM KCl, 5 mM MgCl$_2$, 0.2 mM EDTA, 10% Glycerol, 0.1% NP-40 and 3 mM β-mercaptoethanol), and then eluted with 50 μl 200 ng/μl 3×FLAG peptide (Sigma, F4799) for 30 min at 4 °C twice or with RIPA buffer (50 mM Tris–HCl pH 7.4, 150 mM NaCl, 0.25% sodium deoxycholate, 0.1% NP-40, and 0.1% Triton X-100) for 15 min at 55 °C for 3×FLAG-JMJD3/UTX and NIPBL, respectively. For western blotting, eluates were mixed with loading buffer and boiled. For MS, eluates were subjected to MS analysis at FitGene BioTechnology Co. Ltd. Primary antibodies used were: anti-FLAG (Sigma, F1804), anti-SOX2 (R&D, MAB2018), anti-KLF4 (R&D, AF3158), anti-OCT3/4 (Santa Cruz, sc-8628), anti-c-MYC (R&D, AF3696), anti-NIPBL (BETHYL, A301-779A), anti-MED1 (BETHYL, A300-793), anti-MED12 (BETHYL, A300-774), anti-SMC1A (BETHYL, A300-055), anti-CDK9 (Santa Cruz, sc-484), anti-p300 (Santa Cruz, sc-585), anti-BRD4 (BETHYL, A301-985A100), anti-RBBP5 (BETHYL, A300-109A), anti-RING1B (Cell Signaling Technology, 5694S), anti-H3K27me3 (Millipore, 07-449), anti-Histone H3 (Abcam, ab1791), anti-ACTIN (Sigma, A2066), anti-NANOG (BETHYL A300-397A), and anti-E-cadherin (BD Biosciences, 610181). The dilution for the primary antibodies was 1/1000 for western blotting except for histone H3 (1/3000).

**Flow cytometry analysis.** For the GFP-reporter or tdTomato-reporter, reprogramming cells were digested to single cells and suspended in flow cytometry buffer (2% FBS in PBS) for further analysis with an Accuri C6 Plus flow cytometer (BD Biosciences) and BD Accuri C6 Plus software (v1.0.23.1). Cell-surface marker CD44/ICAM1 profiling was performed as follows: digested cells were suspended in 100 μl flow cytometry buffer containing ~$5 \times 10^5$ cells; staining was carried out at 4 °C for 30 min avoiding light and followed by washing with FACS buffer for three times; data were generated with a LSRFortessa flow cytometer (BD Biosciences) and BD FACSDiva software (v8.0.1). All data were analyzed using FlowJo software (v10.4). Antibodies used were CD44-APC (eBiosciences, 17-0441; 1/200) and ICAM1-PE (eBiosciences, 12-0542; 1/200).

**Sucrose gradient ultracentrifugation.** Nuclear extracts (prepared as above) were fractioned on a 10–30% sucrose gradient by centrifugation using an OPTINMA L-100XP rotor (Beckman) at $25,000 \times g$ at 4 °C for 16 h. Fractions were analyzed by western blotting with the indicated antibodies.

**RNA-seq and analysis.** RNA was extracted as above, and library construction and sequencing were performed at Guangzhou RiboBio Co. Ltd. with an Illumina HiSeq 2500 sequencer. RNA-seq analysis was performed as described[63]. Briefly, reads from the RNA-seq data were aligned to the Ensembl v76 (mm10) transcript annotations using bowtie2 (v2.4.1) and RSEM (v1.2.18). Tag counts were normalized for GC content using EDASeq (v2.0.0). Significantly differential transcript expression was determined using DESeq2 (v1.8.1)[64] ($q$ value < 0.1 and fold change >1.5 in reprogramming samples, and fold change > 2 in MEF samples) and genes are listed in Supplementary Data 3. GO analysis was performed using Goseq (v1.20.0)[65].

**ChIP-qPCR and ChIP-seq.** For H3K27me3 ChIP-seq, *Drosophila* S2 cells were included as a spike-in. Briefly, $1 \times 10^7$ mouse reprogramming and $5 \times 10^6$ *Drosophila* S2 cells were individually cross-linked with 1% formaldehyde for 10 min at room temperature, and quenched by glycine at a final concentration of 0.125 M. Cross-linked cells were rinsed twice with cold PBS and then lysed in ChIP lysis buffer A (50 mM HEPES–KOH pH 7.5, 140 mM NaCl, 1 mM EDTA pH 8.0, 10% glycerol, 0.5% NP-40, 0.25% Triton X-100, and protease inhibitor cocktail) for 10 min at 4 °C. Samples were centrifuged at $1400 \times g$ for 5 min at 4 °C and the pellets re-suspended with ChIP lysis buffer B (1% SDS, 50 mM Tris–HCl pH 8.0, 10 mM EDTA and protease inhibitor cocktail) were lysed for 10 min at 4 °C. Then, lysates of mouse reprogramming and *Drosophila* S2 cells were mixed at a 2:1 ratio. Mixed lysates were sonicated into 150–300 bp fragments using a Bioruptor (Diagenode) sonicator and centrifuged at $14,500 \times g$ at 4 °C for 10 min. To reduce the concentration of SDS, for ChIP-seq, supernatants were diluted twice with ChIP dilution buffer (0.01% SDS, 1.1% Triton X-100, 1.2 mM EDTA, 16.7 mM Tris–HCl, 167 mM NaCl, and protease inhibitor cocktail) and dialyzed with Slide-A-Lyzer Dialysis cassette (Thermo Scientific, 66380); for ChIP-qPCR, the supernatant was

diluted ten times with ChIP immunoprecipitation buffer (0.01% SDS, 1% Triton X-100, 2 mM EDTA, 50 mM Tris–HCl pH 8.0, 150 mM NaCl, and protease inhibitor cocktail). Lysates were incubated with the indicated antibodies overnight at 4 °C and then Protein A/G Dynabeads (Invitrogen) were added to capture the immunoprecipitates. Beads were washed once with low salt buffer (0.1% SDS, 1% Triton X-100, 2 mM EDTA, 20 mM Tris–HCl pH 8.0, and 150 mM NaCl), once with high salt buffer (0.1% SDS, 1% Triton X-100, 2 mM EDTA, 20 mM Tris–HCl pH 8.0, and 500 mM NaCl), once with LiCl buffer (0.25 M LiCl, 1% NP-40, 1% sodium deoxycholate, 1 mM EDTA and 10 mM Tris–HCl pH 8.1), and twice with TE buffer (10 mM Tris–HCl and 1 mM EDTA pH 8.0). Washed beads were eluted with fresh elution buffer (50 mM Tris–HCl, pH 8.0, 10 mM EDTA and 1.0% SDS) at 65 °C with vortex for 30 min. Supernatants were incubated at 65 °C for 8–16 h to reverse the crosslinking and release the immunoprecipitated DNA. After incubation with RNase A and proteinase K, DNA was purified with phenol:chloroform extraction and alcohol precipitation and used for qPCR or sent for sequencing at Guangzhou RiboBio Co. Ltd. with an Illumina HiSeq 2500 sequencer. For KLF4 ChIP-qPCR and ChIP-seq, the method is the same as for H3K27me3, but without *Drosophila* S2 cells as spike-in. For JMJD3 ChIP, samples were handled as above except for the crosslinking and lysis buffers. In brief, reprogramming cells with 3×FLAG-tagged JMJD3 overexpression were treated with 2 mM disuccinimidyl glutarate (DSG, Thermo Scientific) for 30 min prior to formaldehyde crosslinking. After treatment with lysis buffer A, cells were resuspended in lysis buffer C (10 mM Tris–HCl, pH 8.0, 200 mM NaCl, 1 mM EDTA, 0.5 mM EGTA, and protease inhibitor cocktail) for 10 min at 4 °C, and then centrifuged at $1400 \times g$ for 5 min at 4 °C to remove the supernatant. Pellets were lysed in lysis buffer D (10 mM Tris–HCl, pH 8.0, 100 mM NaCl, 1 mM EDTA, 0.5 mM EGTA, 0.1% sodium deoxycholate, 0.5% N-lauroylsarcosine, and protease inhibitor cocktail) for 15 min and then sonicated. Sonicated lysates were diluted (1/3) with RIPA buffer (10 mM Tris–HCl, pH 8.0, 1 mM EDTA, 140 mM NaCl, 1% Triton X-100, 0.1% SDS, 0.1% sodium deoxycholate and protease inhibitor cocktail) for further immunoprecipitation. Primers for ChIP-qPCR are listed in Supplementary Data 2. Antibodies used for ChIP are as follows: anti-H3K27me3 (Millipore, 07-449), anti-H3K27ac (Abcam, ab4729), anti-Pol II (Santa Cruz, sc-899), anti-NIPBL (BETHYL, A301-779A), anti-MED1 (BETHYL, A300-793), anti-MED12 (BETHYL, A300-774), anti-SMC1A (BETHYL, A300-055), anti-CDK9 (Santa Cruz, sc-484), anti-KLF4 (R&D, AF3158), anti-FLAG (Sigma, F1804), mouse anti-IgG (Beyotime, A7028), goat anti-IgG (Beyotime, A7007), and rabbit anti-IgG (Beyotime, A7016). The amount of antibodies for lysates from $1 \times 10^7$ cells was: 5 μg for H3K27me3, H3K27ac and control IgG, and 10 μg for other factors and control IgG.

**ChIP-seq analysis.** Mouse reads from ChIP-seq data for H3K27me3, H3K27ac, KLF4, and JMJD3 were aligned to the mm10 genome using bowtie2 (v2.4.1), and *Drosophila* reads for H3K27me3 were aligned to *Drosophila* dm3 genome. Mouse reads for H3K27me3 were normalized based on the number of mapped *Drosophila* reads ($NF = 2 \times 10^7$/*Drosophila* reads)[32]. Detailed information for the ChIP-seq data is shown in Supplementary Data 4. Peaks were called using DFilter (v1.6)[66] for H3K27me3/ac, and MACS2 (v2.2.5)[67] for KLF4 and JMJD3. The ChIP-seq pileup heatmaps were generated using the glglob.chip_seq_cluster_heatmap function in glbase[68], which collapses all peaks into a non-redundant list and merges all peaks within 400 bp of each other to generate a unique list of peaks before generating sequence read density heatmaps. Pileups were generated using the glbase flat_track. pileup method. The ChIP-seq correlation heatmap was drawn using the glbase function glglob.compare. Motif discovery was performed using HOMER (4.10.3)[69]. All other analyses were also performed using glbase. Definition of ESC-specific enhancers was taken from a previous publication[33]. Other sequencing data used in this study were downloaded from the Gene Expression Omnibus (GEO) database and reprocessed using the pipeline described in this study.

**ATAC-seq and analysis.** A total of 50,000 cells were washed once with cold PBS and re-suspended in 50 μl lysis buffer (10 mM Tris–HCl pH 7.4, 10 mM NaCl, 3 mM MgCl$_2$, and 0.1% IGEPAL CA-630). The suspension was then centrifuged at $500 \times g$ for 10 min at 4 °C, followed by addition of 50 μl transposition reaction mix of TruePrep DNA Library Prep Kit (Vazyme, TD502). Samples were then incubated at 37 °C for 30 min. Transposition reactions were cleaned up using a MinElute PCR Purification Kit (QIAGEN, 28004). ATAC-seq libraries were subjected to five cycles for pre-amplification and amplified by PCR for an appropriate number of cycles. The amplified libraries were purified with a QIAquick PCR Purification Kit (QIAGEN, 28104). Library concentration was measured using VAHTSTM Library Quantification Kit (Vazyme, NQ101). Libraries were sequenced by Vazyme Biotech., Ltd. All sequencing data were mapped onto the mm10 mouse genome assembly using bowtie2 (v2.4.1) (–very-sensitive). Low quality mapped reads were removed using samtools (v1.9) (view –q 35) and only unique reads mapping to a single genomic location and strand were kept. We removed mitochondrial sequences using grep –v chrM (v2.20).

**3C-PCR.** 3C was performed as below[70]. Cells were cross-linked with 2% formaldehyde for 10 min at room temperature and quenched by glycine at a final concentration of 0.125 M. $2 \times 10^6$ cells were lysed with cell lysis buffer (10 mM

Tris–HCl pH 8.0, 10 mM NaCl, 0.2% NP-40 and protease inhibitor cocktail), incubated for 90 min at 4 °C, and then centrifuged at 600×g for 15 min. Nuclei were resuspended in 100 µl of 1× Restriction Enzyme Buffer with 0.3% SDS and incubated at 37 °C for 1 h. Then 20% Triton X-100 was added (the final concentration was 1.8%) to sequester the SDS, and the samples were incubated at 37 °C for 1 h. Each sample was digested with 600 U of restriction enzyme (BamHI and BglII for *Sall1* and *Tdh*; BamHI for *Sall4*) at 37 °C overnight with gentle rotation and the reaction stopped by adding 10% SDS (the final concentration is 1.6%) and further incubation at 65 °C for 20 min. Chromatin DNA was diluted with 1× ligation buffer (30 mM Tris–HCl pH 8.0, 10 mM MgCl₂, 10 mM DTT, and 1 mM ATP) in the presence of 1% Triton X-100 and incubated at 37 °C for 1 h with gentle shaking. 2 µg DNA were ligated with 2000 U T4 ligase and incubated at 16 °C for 4 h and at room temperature for 30 min. After treatment with proteinase K (the final concentration 100 µg/ml) at 65 °C overnight and RNase A (the final concentration 0.4 µg/ml) at 37 °C for 30 min, DNA was purified with phenol/chloroform extraction and ethanol precipitation, and used for PCR amplification of the ligated DNA products.

**Statistical analysis**. It was performed using a two-tailed unpaired Student's *t*-test, ANOVA followed by Holm–Sidak multiple comparison test, or ANOVA followed by Dunnett multiple comparison test (*$P < 0.05$; **$P < 0.01$; ***$P < 0.001$). Values are shown as the mean ± standard error of mean (s.e.m.) and were analyzed with Microsoft Excel 2010 and GraphPad Prism 8 from multiple independent experiments. Detailed *n* values for each panel in the figures are stated in the corresponding legends.

**Reporting summary**. Further information on research design is available in the Nature Research Reporting Summary linked to this article.

## Data availability

RNA-seq, ATAC-seq and ChIP-seq data are available in GEO database under the accession number GSE75005. Published data utilized in this study are under the accession numbers: GSE93027 (RNA-seq for PSCs including ESCs and iPSCs). GSE22562 (MED1, MED12, NIPBL, SMC1A, and SMC3 in ESCs), GSE56098 (p300 in ESCs), GSE19019 (NR5A2 in ESCs), GSE25409 (PRDM14 in ESCs), GSE11431 (ESRRB and STAT3 in ESCs), GSE90895 (SOX2, KLF4, OCT4, and c-MYC in ESCs; H3K27me3, H3K27ac, and H3K9me3 in MEFs, OSKM reprogramming at 48 h, pre-iPSCs and ESCs), GSE44286 (CDK9 in ESCs), GSE67944 (BRD4 in ESCs), GSE106525 (WGBS in MEFs and iPSCs), GSE112520 (WGBS in ESCs) and GSE56986 (WGBS in ESCs). The gating strategies for all flow cytometry experiments are provided in Supplementary Figs. 12–15. A Reporting Summary for this article is available as a Supplementary Information file. All other data supporting the findings of this study are available from the corresponding authors upon reasonable request. Source data are provided with this paper.

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

## Acknowledgements

We thank Dr. Yangming Wang from Peking University for providing the Rex1GFPd2 reporter ESCs, Dr. Jianguo He from Sun Yat-sen University for providing the *Drosophila* S2 cell line, and Dr. Yaoyao Wu and Yongqiang Chen from GIBH for technical assistance. We also thank all members of the Qin, Esteban, and Pei laboratories for their support. This work is supported by the National Key Research and Development Program of China (2016YFA0100300 to B.Q.; 2016YFA0100102 and 2018YFA0106903 to M. A.E.; 2018YFA0106902 to J.H.), the Strategic Priority Research Program of the Chinese Academy of Sciences (XDA16030502 to M.A.E.), National Natural Science Foundation of China (31671537 to M.A.E.; 31970589 to A.P.H.; 31950410553 to C.W.; 31850410486 to I.A.B.; 31771454 to R.J.), the Innovative Team Program of Guangzhou Regenerative Medicine and Health Guangdong Laboratory (2018GZR110103001 to M.A.E. and B.Q.), the Guangdong Province Science and Technology Program (2015A030308007 to B.Q.; 2016B030229007 and 2017B050506007 to M.A.E.; 2019A050510004 to A.P.H.), the Guangzhou Science and Technology Program (201907010039 to B.Q. and 201807010066 to M.A.E.), the Shenzhen Municipal Commission of Science and Technology Innovation (ZDSYS20190902093401689 and JCYJ20180507182044945 to B.L.). The Guangdong Provincial Key Laboratory of Stem Cells and Regenerative Medicine is supported by the Science and Technology Planning Project of Guangdong Province, China (2017B030314056). B.Q. is supported by the Guangdong Special Support Program (2016TX03R366). C.W. is supported by a Chinese Academy of Sciences President's International Fellowship Initiative for Postdoctoral Researchers (2019PB0177). G.V. is supported by a Chinese Academy of Sciences President's International Fellowship for Foreign Experts (2020FSB0002).

## Author contributions

B.Q. conceived the idea and Y.H., H. Zhang and M.A.E. contributed to the idea. B.Q. and M.A.E. supervised the study. Y.H., H. Zhang and C.T. conducted most of the experiments. A.P.H. conducted most of the bioinformatics analysis, and Y.H., Z.Y., I.A.B., X. Wu, and Y. Lai contributed to the analysis. L.W., X.Q., Y.T., X.X., R.L., J.S., X. Wu, M.P., L.S. and T.A. contributed to the experiments. B.Q., Y.H., H. Zhang, and M.A.E. analyzed the data. T.S., S.A., G.T., Jinlong C., H. Zheng, L.G., X. Wen, Y. Li, X.H., J.H., and Jiekai C. provided technical assistance or reagents. L.L., G.P., D.P., and M.A.E. provided infrastructural support. G.J., C.W., G.V., F.G., X.B., B.L., G.P., V.M., and R.J. provided relevant advice. B.Q., Y.H., H. Zhang, and M.A.E. wrote the manuscript, and B.Q. and M. A.E. approved the final version. B.Q. and M.A.E. provided most of the financial support.

## Competing interests

The authors declare no competing interests.
