## [Peer Review File · Nature Communications]

Reviewers' comments:

Reviewer #1 (Remarks to the Author):

Huang et al., find that JMJD3 facilitates reprogramming to pluripotency in a context-dependent manner. Specifically, gain-of-function of JMJD3 boost reprogramming of early passage MEFs, while it is detrimental in late passage fibroblasts. The authors also made a big effort to delineate the mechanistic basis for the observed phenotypes and provide evidence that JMJD3 cooperates with Klf4 to enhance iPSC reprogramming efficiency. Overall, I feel that the study is well done and provides a number of observations that are of interest for the reprogramming and stem cell field. Yet, I have some points that the authors should address before the manuscript is ready for publication:

Major points

1. Most of the genome-wide analysis in the manuscript are performed in MEFs undergoing reprogramming at day 5 and 10, failing to address the direct effect of JMJD3 on the gene regulatory network of MEFs. Thus, it would be interesting to see how JMJD3 gain- and loss-of-function affect the transcriptome and H3K27me3 levels (i.e. RNA-seq and ChIP-seq) of MEFs at P2 and P4. The authors should also include iPSCs as a control for both RNA-seq and ChIP-seq experiments.
2. Considering the previous observation that JMJD3 is a negative regulator of iPS reprogramming (PMID: 23452852), the authors should try to expand and strengthen their observations by testing the effect of JMJD3 overexpression and knockdown in a different somatic cell type. For example, they could see how JMJD3 levels affect the reprogramming of neural progenitor cells which already express KLF4 and reprogram at high efficiency.
3. Given that some sites associated with JMJD3 binding are enriched for genes related to neural differentiation and other developmental pathways, it would be interesting to test how manipulation of JMJD3 levels affect iPSC differentiation (three lineage differentiation in EBs) and transdifferentiation of MEFs in alternative fates (MEF to neurons or cardiomyocytes).
4. For the EpiSCs to naïve ESCs reprogramming, the authors should perform flow cytometric analysis to quantify the reprogramming efficiency and use appropriate reporter lines (e.g., Rex1-GFP).
5. JMJD3 enhances the reprogramming of OKS infected MEFs (without Myc) in iCD1 medium. Does it also enhance reprogramming of OSK MEFs in FBS+LIF culture conditions?

Minor points:

1. I do not find clear the heatmap in Figure 3A, a PCA representation would be probably more informative. Again, the paper should include here MEF and iPSCs as controls.

Reviewer #2 (Remarks to the Author):

JMJD3 facilitates the architectural role of KLF4 and enhances somatic cell reprogramming in a senescence-dependent manner

Summary

This study clearly demonstrates the important role JMJD3 plays with KLF4 in the mouse reprogramming process. I recommend attention to the following major and minor issues which can be addressed by modification to the text.

Major

1. As all work was done in mouse cells, "mouse" needs to be added to the title and abstract to make this point clear. I suggest "JMJD3 facilitates the architectural role of KLF4 and enhances mouse somatic cell reprogramming in a senescence-dependent manner".
2. On line 221 the authors state, "Therefore, endogenous JMJD3 is also necessary for reprogramming and its mechanism of action is distinct from UTX." This statement is overstating the results. Their data show that JMJD3 is important for efficient reprogramming but not that it is necessary as reprogrammed cells were obtained from JMJD3 knock out cells. Any other similarly overstated language should be addressed.
3. On line 286 the authors state, "Therefore, changes in H3K27me3 induced by JMJD3 overexpression in reprogramming happen at all genic regions." Based on the data they show this is overstated. It is clear that there is an overall genome-wide trend to reduced modification. But the authors have not shown that this occurs at all regions. To show this they would need to display the range of change in peak intensity for all regions and show that this range only displays a loss and no gain at any regions with H3K27me3. I do not feel that this is required in their analysis and it would be appropriate to simply address the overstatement by modifying the text.

Minor

1. Please define 4F more clearly, or consider using OSKM rather than 4F.
2. In Fig1A and B the y axis needs a better title that declares what is being measured.
3. Line 194: please reword "This abolished the reducing effect of exogenous JMJD3 on H3K27me3 levels". This is confusing. I suggest something like "This abolished the effect exogenous JMJD3 had on reducing H3K27me3 levels"
4. Line 284: Notably, we detected substantial H3K27me3 enrichment at ESC-specific enhancers⁴⁰ in both cell types too, and H3K27me3 levels at these regions were potently reduced by exogenous JMJD3. Remove "too".

Reviewer #3 (Remarks to the Author):

In this study, Huang et al. examine the impact of ectopically expressing JMJD3 during OSKM-induced reprogramming of MEFs. They find that exogenous JMJD3 expression induces the upregulation of Ink4a and degradation of PHF20 in a reprogramming independent manner. These two events were previously reported to inhibit reprogramming as recapitulated in this study using late passage (P4) MEFs. In contrast, JMJD3 is uncovered as a potent positive regulator of reprogramming when ectopically expressed in early passage (P2) MEFs. In this context, exogenous JMJD3 is shown to enhance the frequency of fully reprogrammed cells formed, and this effect is dependent on JMJD3's catalytic activity. Accordingly, the authors report an accelerated loss of H3K27me3 at target genes that are normally upregulated during reprogramming. Interestingly, they show that JMJD3 recruitment is dependent on KLF4 and furthermore that exogenous JMJD3 interacts with essential constituents of enhancer protein complexes including Mediator, Cohesin and NIPBL to form part of activating protein complexes at functionally relevant enhancers. While this observation is novel and interesting, the authors do not explore whether JMJD3 is essential for enhancer-promoter looping and/or the mechanism by which JMJD3 mediates its action at these sites upon reprogramming and/or in fully reprogrammed cells.

Specific comments:

1- The authors find that ectopically expression JMJD3 similarly enhances Ink4a expression in early and late passage MEFs. However, the starting level of Ink4a expression is higher in the latter. Is this also correlating with higher endogenous Jmjd3 expression in these cells? Please add the expression profiles of Jmjd3 and Utx in Figure 1a.

2- While ectopically expressing JMJD3 reduces the formation of AP+ (early) reprogrammed cells, it favours the formation of GFP+ (fully) reprogrammed cells, implying a role for JMJD3 during late stages of reprogramming. Could this be directly examining using high-resolution cell surface markers and flow cytometry analysis as previously described (O' Malley et al. Nature 2013)?

3- Figure 2a - different timing of reprogramming (day 5 and day 10) are used in this study. Could the authors also quantify the loss of global H3K27me3 in a catalytic-dependent manner at these two timings to corroborate an early and/or late action of JMJD3? Please note that replicates are required for Figure 2a, c and h as well as statistical analysis. It would also be informative to compare the level of H3K27me3 in pre-IPSCs and fully reprogrammed cells in absence of exogenous JMJD3 to validate whether this epigenetic remodelling is normally occurring during this transition.

4- Figure 2e-g – loss of endogenous JMJD3 expression is shown to inhibit reprogramming in contrast to previous reports (e.g. Zhao W et al. Cell 2013). Does the discrepancy relate to a passage number effect? In other words, would they obtain different results if using late passage (P4) MEFs? Could the authors also show the % of AP+ colonies to evaluate whether JMJD3 knock-down in their system would enhance the formation of early reprogrammed cells, but impedes the formation of fully reprogrammed cells only? In the same line of thought, is JMJD3 required for the reprogramming or the maintenance of fully reprogrammed cells? Ideally, the authors could use a conditional rescued system of JMJD3KO MEFs to further explore the exact stage(s) of reprogramming that is impacted by JMJD3 action and associated effects in terms of chromatin changes and gene expression.

5- Figure 3a-c show the enhancement and/or inhibition of gene expression upon ectopic expression or knock-down of JMJD3 in MEFs upon reprogramming. Do these gene expression changes not simply reflect differences in reprogramming efficiency rather than a direct effect of JMJD3 gain-and-loss of function? This needs to be discussed by the authors.

6- Figure 3d – Loss of H3K27me3 seems to happen faster when ectopically expressing JMJD3 in a WT compared to JMJD3KO background. Could the authors discuss this difference? Again, it would be informative to also examine the profile of H3K27me3 in pre-IPSC and fully reprogramming cells (endogenous Jmjd3 action).

7- Figure 3e – could the authors show the level of H3K27me3 in the same populations at desert regions to access whether the effect is loci specific as opposed to genome-wide?

8- Figure 3f and Supplementary Fig. 4d,e – could the authors separately look at H3K27me3 changes upon reprogramming at the two distinct sets of loci: (1) loci up-regulated from day 5 (e.g. Cdh1, Sall1 and Sall4) and (2) loci only up-regulated later on at day 10 of reprogramming (e.g. Esrrb, Zfp42 and Dppa5a)? It would be interesting indeed to more precisely delineate whether (or not) the latter show a loss of H3K27me3 at their TSS and/or enhancers prior to gene activation. As the present, the authors have very little evidence supporting their statement lines 296-300 that “ for many pluripotency loci the removal of H3K27me3 in the early phase of reprogramming primes them for transcriptional activation in the late phase”. As further evidence, could the authors also corroborate the kinetics of H3K27me3 loss and acquisition of H3K27ac at the two sets of loci? Additionally, it might be interesting to explore whether other genomic and epigenetic features underline the observed difference in the

gene activation timing of these sets of loci (e.g. CpG low/intermediate/high TSSs, DNA methylation and H3K9me3 status at TSS and/or enhancers in MEFs). In the same line of thought, could the authors repeat their motif analysis shown in Supplementary Fig. 4g separately for the two sets of loci?

9- Figure 4l - how cluster 1, 2 and 3 correlates with the above sets of loci? Please also show GO analysis for the three clusters in Supplementary Fig. 6i. Line 387, could the authors acknowledge the possibility that JMJD3 might be recruited at loci within cluster 2 by other transcription factors that KLF4 upon reprogramming?

10- Figure 5 - the authors interestingly uncover that JMJD3 form part of an activating protein complexes at enhancers. However, they do not explore whether JMJD3 is essential for the activity of enhancers and the mechanisms by which JMJD3 mediates its action upon reprogramming and/or in fully reprogrammed cells. Could the authors make use of their rescued JMJD3 KO MEFs with WT JMJD3 and mutated forms to compare the profile of enhancer protein binding and enhancer-promoter interactions in fully reprogrammed clones? Any effort in that direction would be a great addition to this study.

Other minor comments:

1- Line 180 - JMJD3 does not promote reprogramming in a Vc-dependent manner as stated. However, its action seems to be enhanced in the presence of Vc.

2- Figure 5h - there is no indication of the reprogramming timing used in these experiments. Same comment for Supplementary Fig. 8c,e.

3- Supplementary Fig.1j - replicates required as well as statistical analysis.

4- Supplementary Fig.2j - could the authors provide a quantification of GFP+ and NANOG+ colonies in the same experiments?

5- Supplementary Fig. 5b, e and g - triplicate experiments should be best provided.

6- Supplementary Fig. 6h - should the authors confidently use data generated from other reprogramming studies (e.g. Chronis et al.) using different protocols and MEF passage numbers?

7- Supplementary Fig. 7d. Please provide information about datasets used in this figure.

Reviewer #1 Comments:

Huang et al., find that JMJD3 facilitates reprogramming to pluripotency in a context-dependent manner. Specifically, gain-of-function of JMJD3 boost reprogramming of early passage MEFs, while it is detrimental in late passage fibroblasts. The authors also made a big effort to delineate the mechanistic basis for the observed phenotypes and provide evidence that JMJD3 cooperates with Klf4 to enhance iPSC reprogramming efficiency. Overall, I feel that the study is well done and provides a number of observations that are of interest for the reprogramming and stem cell field. Yet, I have some points that the authors should address before the manuscript is ready for publication:

Major points

1. Most of the genome-wide analysis in the manuscript are performed in MEFs undergoing reprogramming at day 5 and 10, failing to address the direct effect of JMJD3 on the gene regulatory network of MEFs. Thus, it would be interesting to see how JMJD3 gain- and loss-of-function affect the transcriptome and H3K27me3 levels (i.e. RNA-seq and ChIP-seq) of MEFs at P2 and P4. The authors should also include iPSCs as a control for both RNA-seq and ChIP-seq experiments.
2. Considering the previous that JMJD3 is a negative regulator of iPS reprogramming (PMID: 23452852), the authors should try to expand and strengthen their observations by testing the effect of JMJD3 overexpression and knockdown in a different somatic cell type. For example, they could see how JMJD3 levels affect the reprogramming of neural progenitor cells which already express KLF4 and reprogram at high efficiency.
3. Given that some sites associated with JMJD3 binding are enriched for genes related to neural differentiation and other developmental pathways, it would be interesting to test how manipulation of JMJD3 levels affect iPSC differentiation (three lineage differentiation in EBs) and transdifferentiation of MEFs in alternative fates (MEF to neurons or cardiomyocytes).

4. For the EpiSCs to naïve ESCs reprogramming, the authors should perform flow cytometric analysis to quantify the reprogramming efficiency and use appropriate reporter lines (e.g., Rex1-GFP).

5. JMJD3 enhances the reprogramming of OKS infected MEFs (without Myc) in iCD1 medium. Does it also enhance reprogramming of OSK MEFs in FBS+LIF culture conditions?

Minor points:

1. I do not find clear the heatmap in Figure 3A, a PCA representation would be probably more informative. Again, the paper should include here MEF and iPSCs as controls.

Our responses:

“Overall, I feel that the study is well done and provides a number of observations that are of interest for the reprogramming and stem cell field. ...”

Response: We thank the reviewer for the positive assessment of our manuscript and the helpful comments.

Please note that we would like to change the title to a new one: “**JMJD3 acts in tandem with KLF4 to facilitate reprogramming to pluripotency**”. We believe this title is more concise and with greater perspective. We hope the reviewer will agree with the change. Of course, if the reviewer and/or editor think it is not appropriate, we can change to: “JMJD3 facilitates the architectural role of KLF4 and enhances mouse somatic cell reprogramming in a senescence dependent manner”.

Major points

1-“Most of the genome-wide analysis in the manuscript are performed in MEFs undergoing reprogramming at day 5 and 10, failing to address the direct effect of JMJD3 on the gene regulatory network of MEFs. Thus, it would be interesting to see how JMJD3 gain- and loss-of-function affect the transcriptome and H3K27me3 levels (i.e. RNA-seq and CHIP-seq) of MEFs at

P2 and P4. The authors should also include iPSCs as a control for both RNA-seq and ChIP-seq experiments.”

Response: The reviewer raised a relevant point. To address this, we have added new RNA-seq for:

- (1) OG2 MEFs at P2 and P4, with or without JMJD3 overexpression (OE).
- (2) *Jmjd3* wild-type (WT) & knockout (KO) MEFs at P2 and P4.

Basically, the conclusion is that JMJD3 regulates distinct sets of genes in MEFs compared with reprogramming. The results are discussed in page 11 line 271 in the revised manuscript.

Specifically, for (1) OG2 MEFs, JMJD3 OE changed the expression levels of 672 genes at P2 and approximately half that number (323) at P4 (fold change >2; **REBUTTAL Fig. 1A**). Gene ontology (GO) analysis showed that JMJD3 OE-regulated genes are mainly related to cell cycle, in perfect correlation with the well-known effect of JMJD3 on *Ink4a* (**REBUTTAL Fig. 1B**). For (2) *Jmjd3* WT and KO MEFs, because the original MEFs used in our study were limited (MEFs were from Shizuo Akira and Giuseppe Testa), we have now generated conditional *Jmjd3* KO mice producing a deletion of exons 14-20 (encompassing the JmjC domain)¹ in an OG2 background (OG2-*Jmjd3*^{fl/fl}). To achieve the KO, we delivered Cre with adeno-associated viruses into MEFs. Please see page 10 line 237 in the revised text and **NEW Supplementary Figures 2m, n** for confirmation of the expected phenotype (compared to the other KO MEFs) in these MEFs.

In the RNA-seq of these conditional KO (cKO) MEFs, *Jmjd3* deficiency induced few transcriptional changes compared with JMJD3 OE, and these were largely distinct between P2 and P4 (**REBUTTAL Fig. 1C**). In contrast to JMJD3 OE, *Jmjd3* cKO did not change much *Ink4a* either (**REBUTTAL Fig. 1D**). The correlation between genes upregulated with JMJD3 OE (OE-Up) and genes downregulated with *Jmjd3* cKO (cKO-Down) was also modest (**REBUTTAL Fig. 1E**).

The above-mentioned analyses indicate that exogenous and endogenous JMJD3 largely regulate distinct sets of genes in MEFs in basal conditions, with the endogenous role being more modest as expected.

In addition, we correlated the genes upregulated upon JMJD3 OE in OSKM reprogramming at day 5 (OE reprogramming-Up) and the genes reduced by *Jmjd3* knockdown in the same setting (KD reprogramming-Down), with the new data in P2 MEFs OE-Up and cKO-Down, respectively. We found that ~16% (52 genes) of OE reprogramming-Up and only ~0.5% (4 genes) of KD reprogramming-Down genes overlap with OE-Up and cKO-Down in MEFs, respectively (**REBUTTAL Fig. 1F** and **NEW Supplementary Fig. 3d**). We have now included a heatmap containing all these datasets in **NEW Figure 3a** (**REBUTTAL Fig. 1G**), which contains iPSCs and ESCs too. The majority of the overlapping genes are MEF-enriched and transiently induced in reprogramming (**REBUTTAL Fig. 1F**). For the few overlapping pluripotent stem cell (PSC)-enriched genes (*Clo18a1*, *Krt18*, *Epas1*, *Icam1*, *Igfbp2*, *Notch3*, and *Jam2*) between OE reprogramming-Up at day 5 and OE-up in MEFs (**REBUTTAL Fig. 1F**), only *Epas1* (encoding HIF2 α) has been reported to have a function in either PSCs or reprogramming. HIF2 α is beneficial for the early phase of human cell reprogramming by promoting glycolysis and detrimental for the late phase by inducing TNF-related apoptosis-inducing ligand (TRAIL)². However, we did not find significant expression changes of the glycolytic and TRAIL target genes (data not shown) upon JMJD3 OE in early and late reprogramming respectively. So, the relevance of HIF2 α regulation by JMJD3 in reprogramming is still unclear and we opted for not discussing it.

Based on these results, we conclude that the effect of modulating JMJD3 in MEFs and reprogramming has different functional consequences. Accordingly, we think it is not necessary to perform ChIP-seq for H3K27me3 in MEFs with JMJD3 modulation. We hope that the reviewer will agree with us.

2-“Considering the previous observation that JMJD3 is a negative regulator of iPS reprogramming (PMID: 23452852), the authors should try to expand and strengthen their observations by testing the effect of JMJD3 overexpression

and knockdown in a different somatic cell type. For example, they could see how JMJD3 levels affect the reprogramming of neural progenitor cells which already express KLF4 and reprogram at high efficiency.”

Response: The reviewer raised an interesting point. In our hands, reprogramming of neural progenitor cells (NPCs) with OKM was relatively efficient, producing 30-40 iPSC colonies at day 10 with Vc (**REBUTTAL Fig. 2A**). However, we could not produce iPSC colonies with OSM in the same time frame, in agreement with an earlier report³ (Supplementary Table 1 of Kim *et al. Nature* 2008, shown here as **REBUTTAL Fig. 2B**). We also noticed previous reports describing that JMJD3 regulates NPC to neuron differentiation⁴⁻⁶, and our RT-qPCR confirmed that JMJD3 OE in NPCs induces neuronal markers (**REBUTTAL Fig. 2C**). Because of all this, we adjusted the reprogramming procedure (JMJD3 or sh*Jmjd3* virus transduction was delayed 2 days after OKM delivery), and we could observe enhanced and reduced reprogramming efficiency by JMJD3 OE and knockdown, respectively, as in MEFs with OSK/M (**REBUTTAL Fig. 2A**).

In conclusion, JMJD3 promotes reprogramming in cell types other than fibroblasts. We have now included this information into the revised manuscript (see **NEW Supplementary Fig. 2o** and page 10 line 244).

3-“Given that some sites associated with JMJD3 binding are enriched for genes related to neural differentiation and other developmental pathways, it would be interesting to test how manipulation of JMJD3 levels affect iPSC differentiation (three lineage differentiation in EBs) and transdifferentiation of MEFs in alternative fates (MEF to neurons or cardiomyocytes).”

Response: According to the literature, JMJD3 regulates ESC differentiation towards the three germ lineages⁷, including neural differentiation from pluripotent stem cells and NPCs^{8,9}. We believe that iPSCs should behave the same as ESCs, so we have not repeated this work here but have added a note explaining all this in the revised manuscript (see page 18 line 471).

As for transdifferentiation of MEFs into other cell types, we tested neuronal transdifferentiation. As shown in **REBUTTAL Figure 3A**, a 3-day induction of OSKM in P2 Dox-inducible secondary reprogrammable MEFs¹⁰ in mouse ESC medium (without LIF) with Vc, which prevents the generation of

iPSCs and is sufficient for neuronal transdifferentiation^{11,12}, followed by a switch to neuronal medium for 14 days could successfully transdifferentiate MEFs into TUJ1⁺ neurons, albeit with low efficiency (**REBUTTAL Fig. 3B**). Exogenous JMJD3 significantly enhanced this conversion, as detected both by immunofluorescence and RT-qPCR (**REBUTTAL Fig. 3B, C**). These results suggest a role for JMJD3 in promoting neuronal transdifferentiation from MEFs. However, without a neuronal reporter (such as Tau-EGFP) to exclude neuronal cells during MEF preparation¹³, we think these results may not be stringent enough to be included in the revised manuscript. Moreover, although interesting, we think that proper demonstration that JMJD3 promotes transdifferentiation is out of the scope of this work. Accordingly, we also removed some figure panels (previous **Supplementary Fig. 6f, g**) related to this point for simplification of the message and to avoid confusion. We hope that the reviewer will agree with us and to point at this interesting possibility we have included a note in the revised manuscript (page 23 line 627).

4-“For the EpiSCs to naïve ESCs reprogramming, the authors should perform flow cytometric analysis to quantify the reprogramming efficiency and use appropriate reporter lines (e.g., Rex1-GFP).”

Response: The reviewer raised an important point. We have now converted Rex1GFPd2 reporter-ESCs¹⁴ to Epiblast-like cells (EpiLCs) by culturing them in N2B27 medium with bFGF/Activin A and letting them stabilize for 5 passages, at which point GFP fluorescence completely disappeared. Next, we induced Epi-to-naïve transition for both OG2-EpiSCs and Rex1GFPd2-EpiLCs with KLF4 in 2iL+Vc medium, with or without exogenous JMJD3. Then, we performed flow cytometry analysis, which showed that JMJD3 induces an ~3-fold increase in reprogramming efficiency using both reporter cell lines (**REBUTTAL Fig. 4A, B**), a degree similar to our colony counting data (Fig. 4g in our original manuscript). These results confirm that JMJD3 enhances the Epi-to-naïve transition. We have now included these data into the revised manuscript (**NEW Fig. 4g, NEW Supplementary Fig. 6i** and page 16 line 429 and 430).

5-“JMJD3 enhances the reprogramming of OKS infected MEFs (without Myc) in iCD1 medium. Does it also enhance reprogramming of OSK MEFs in FBS+LIF culture conditions?”

Response: The reviewer raised a good point. We have now tested the effect of exogenous JMJD3 on OSK reprogramming in FBS+LIF culture condition with and without Vc, and found a moderate but reproducible and significant (~50%) increase in efficiency (**REBUTTAL Fig. 5**). We have now included the data into the revised manuscript (**NEW Supplementary Fig. 6d** and page 15 line 404).

Minor points

1-“I do not find clear the heatmap in Figure 3A, a PCA representation would be probably more informative. Again, the paper should include here MEF and iPSCs as controls.”

Response: As suggested by the reviewer, we have done PCA with all the conditions including MEFs and iPSCs/ESCs (**REBUTTAL Fig. 6A**). All MEFs cluster together. iPSCs cultured in ESC medium with 2i on feeders, and ESCs cultured in ESC medium on gelatin, cluster more loosely, likely due to the different origins and/or culture conditions. Reprogramming cells at each time point cluster together and separate from both MEFs and iPSCs/ESCs. The difference between with or without JMJD3 OE or knockdown among the reprogramming cells at each time was relatively mild when compared with MEFs and iPSC/ESCs, which is understandable considering the large numbers of genes that change between MEFs and iPSCs/ESCs and the relatively low number of differentially expressed genes between JMJD3 gain or loss (**REBUTTAL Fig. 1G**). Hence, we have opted for not adding the PCA in the revised manuscript, which we hope the reviewer will agree with. Nevertheless, to clarify this point in the main text, we replaced the heatmap of previous Figure 3a with a new one (**REBUTTAL Fig. 1G**), showing genes that are only expressed differentially by JMJD3 modulation in reprogramming, and as requested also included MEFs and iPSCs/ESCs as controls (see page 11 line 283). Similarly, we replaced previous Supplementary Figure 3d with new panels showing the GO analysis for JMJD3-regulated transiently activated (**REBUTTAL Fig. 6B** and **NEW Supplementary Fig. 3e**) and PSC-enriched

genes (**REBUTTAL Fig. 6C** and **NEW Supplementary Fig. 3f**) genes. These data are discussed in the revised manuscript (page 11 line 290).

Reviewer #2 Comments:

JMJD3 facilitates the architectural role of KLF4 and enhances somatic cell reprogramming in a senescence-dependent manner

This study clearly demonstrates the important role JMJD3 plays with KLF4 in the mouse reprogramming process. I recommend attention to the following major and minor issues which can be addressed by modification to the text.

Major

1. As all work was done in mouse cells, “mouse” needs to be added to the title and abstract to make this point clear. I suggest “JMJD3 facilitates the architectural role of KLF4 and enhances mouse somatic cell reprogramming in a senescence-dependent manner”.

2. On line 221 the authors state, “Therefore, endogenous JMJD3 is also necessary for reprogramming and its mechanism of action is distinct from UTX.” This statement is overstating the results. Their data show that JMJD3 is important for efficient reprogramming but not that it is necessary as reprogrammed cells were obtained from JMJD3 knock out cells. Any other similarly overstated language should be addressed.

3. On line 286 the authors state, “Therefore, changes in H3K27me3 induced by JMJD3 overexpression in reprogramming happen at all genic regions.” Based on the data they show this is overstated. It is clear that there is an overall genome-wide trend to reduced modification. But the authors have not shown that this occurs at all regions. To show this they would need to display the range of change in peak intensity for all regions and show that this range only displays a loss and no gain at any regions with H3K27me3. I do not feel that this is required in their analysis and it would be appropriate to simply address the overstatement by modifying the text.

Minor

1. Please define 4F more clearly, or consider using OSKM rather than 4F.

2. In Fig1A and B the y axis needs a better title that declares what is being measured.

3. Line 194: please reword “This abolished the reducing effect of exogenous JMJD3 on H3K27me3 levels”. This is confusing. I suggest something like “This abolished the effect exogenous JMJD3 had on reducing H3K27me3 levels”

4. Line 284: Notably, we detected substantial H3K27me3 enrichment at ESC-specific enhancers⁴⁰ in both cell types too, and H3K27me3 levels at these regions were potently reduced by exogenous JMJD3. Remove “too”.

Our responses:

“This study clearly demonstrates the important role JMJD3 plays with KLF4 in the mouse reprogramming process.”

Response: We thank the reviewer for the positive assessment of our manuscript and the helpful comments.

Major:

1-“As all work was done in mouse cells, ‘mouse’ needs to be added to the title and abstract to make this point clear. I suggest ‘JMJD3 facilitates the architectural role of KLF4 and enhances mouse somatic cell reprogramming in a senescence-dependent manner’.”

Response: We agree with the reviewer and have revised the abstract to emphasize that the study is in mouse (page 3 line 52). As for the title, considering the suggestions by all the reviewers, we would like to change it to a new one: “**JMJD3 acts in tandem with KLF4 to facilitate reprogramming to pluripotency**”. We believe this title is more concise and with greater perspective and would not need “mouse” to be added in. We hope the reviewer will agree with the change. Of course, if the reviewer and/or editor think it is not appropriate, we can change to “JMJD3 facilitates the architectural role of KLF4 and enhances mouse somatic cell reprogramming in a senescence dependent manner”.

2-“On line 221 the authors state, ‘Therefore, endogenous JMJD3 is also necessary for reprogramming and its mechanism of action is distinct from UTX.’ This statement is overstating the results. Their data show that JMJD3 is important for efficient reprogramming but not that it is necessary as reprogrammed cells were obtained from JMJD3 knock out cells. Any other similarly overstated language should be addressed.”

Response: We agree with the reviewer and have revised the text to avoid any overstatements (page 10 line 249).

3-“On line 286 the authors state, “Therefore, changes in H3K27me3 induced by JMJD3 overexpression in reprogramming happen at all genic regions.” Based on the data they show this is overstated. It is clear that there is an overall genome-wide trend to reduced modification. But the authors have not shown that this occurs at all regions. To show this they would need to display the range of change in peak intensity for all regions and show that this range only displays a loss and no gain at any regions with H3K27me3. I do not feel that this is required in their analysis and it would be appropriate to simply address the overstatement by modifying the text.”

Response: We agree with the reviewer and have revised the text to tone down this claim (see page 13 line 343).

Minor

1-“Please define 4F more clearly, or consider using OSKM rather than 4F.”

Response: We have now replaced 4F with OSKM in the revised manuscript.

2-“In Fig1A and B the y axis needs a better title that declares what is being measured.”

Response: We have now revised the title of **NEW Figure 1a, c**.

3-“Line 194: please reword ‘This abolished the reducing effect of exogenous JMJD3 on H3K27me3 levels’. This is confusing. I suggest something like ‘This abolished the effect exogenous JMJD3 had on reducing H3K27me3 levels.’”

Response: We have revised the text as suggested (see page 9 line 208).

4-“Line 284: Notably, we detected substantial H3K27me3 enrichment at ESC-specific enhancers⁴⁰ in both cell types too, and H3K27me3 levels at these regions were potently reduced by exogenous JMJD3. Remove ‘too.’”

Response: We have removed this as suggested.

Reviewer #3 Comments (Remarks to the Author):

In this study, Huang et al. examine the impact of ectopically expressing JMJD3 during OSKM-induced reprogramming of MEFs. They find that exogenous JMJD3 expression induces the upregulation of Ink4a and degradation of PHF20 in a reprogramming independent manner. These two events were previously reported to inhibit reprogramming as recapitulated in this study using late passage (P4) MEFs. In contrast, JMJD3 is uncovered as a potent positive regulator of reprogramming when ectopically expressed in early passage (P2) MEFs. In this context, exogenous JMJD3 is shown to enhance the frequency of fully reprogrammed cells formed, and this effect is dependent on JMJD3's catalytic activity. Accordingly, the authors report an accelerated loss of H3K27me3 at target genes that are normally upregulated during reprogramming. Interestingly, they show that JMJD3 recruitment is dependent on KLF4 and furthermore that exogenous JMJD3 interacts with essential constituents of enhancer protein complexes including Mediator, Cohesin and NIPBL to form part of activating protein complexes at functionally relevant enhancers. While this observation is novel and interesting, the authors do not explore whether JMJD3 is essential for enhancer-promoter looping and/or the mechanism by which JMJD3 mediates its action at these sites upon reprogramming and/or in fully reprogrammed cells.

Specific comments:

1- The authors find that ectopically expression JMJD3 similarly enhances Ink4a expression in early and late passage MEFs. However, the starting level of Ink4a expression is higher in the latter. Is this also correlating with higher endogenous Jmjd3 expression in these cells? Please add the expression profiles of Jmjd3 and Utx in Figure 1a.

2- While ectopically expressing JMJD3 reduces the formation of AP+ (early) reprogrammed cells, it favours the formation of GFP+ (fully) reprogrammed cells, implying a role for JMJD3 during late stages of reprogramming. Could this be directly examining using high-resolution cell surface markers and flow cytometry analysis as previously described (O' Malley et al. Nature 2013)?

3- Figure 2a - different timing of reprogramming (day 5 and day 10) are used in this study. Could the authors also quantify the loss of global H3K27me3 in a catalytic-dependent manner at these two timings to corroborate an early and/or late action of JMJD3? Please note that replicates are required for Figure 2a, c and h as well as statistical analysis. It would also be informative to compare the level of H3K27me3 in pre-IPSCs and fully reprogrammed cells in absence of exogenous JMJD3 to validate whether this epigenetic remodelling is normally occurring during this transition.

4- Figure 2e-g – loss of endogenous JMJD3 expression is shown to inhibit reprogramming in contrast to previous reports (e.g. Zhao W et al. Cell 2013). Does the discrepancy relate to a passage number effect? In other words, would they obtain different results if using late passage (P4) MEFs? Could the authors also show the % of AP+ colonies to evaluate whether JMJD3 knock-down in their system would enhance the formation of early reprogrammed cells, but impedes the formation of fully reprogrammed cells only? In the same line of thought, is JMJD3 required for the reprogramming or the maintenance of fully reprogrammed cells? Ideally, the authors could use a conditional rescued system of JMJD3KO MEFs to further explore the exact stage(s) of reprogramming that is impacted by JMJD3 action and associated effects in terms of chromatin changes and gene expression.

5- Figure 3a-c show the enhancement and/or inhibition of gene expression upon ectopic expression or knock-down of JMJD3 in MEFs upon reprogramming. Do these gene expression changes not simply reflect differences in reprogramming efficiency rather than a direct effect of JMJD3 gain-and-loss of function? This needs to be discussed by the authors.

6- Figure 3d – Loss of H3K27me3 seems to happen faster when ectopically expressing JMJD3 in a WT compared to JMJD3KO background. Could the authors discuss this difference? Again, it would be informative to also examine the profile of H3K27me3 in pre-IPSC and fully reprogramming cells (endogenous Jmjd3 action).

7- Figure 3e – could the authors show the level of H3K27me3 in the same populations at desert regions to assess whether the effect is loci specific as opposed to genome-wide?

8- Figure 3f and Supplementary Fig. 4d,e – could the authors separately look at H3K27me3 changes upon reprogramming at the two distinct sets of loci: (1) loci up-regulated from day 5 (e.g. *Cdh1*, *Sall1* and *Sall4*) and (2) loci only up-regulated later on at day 10 of reprogramming (e.g. *Esrrb*, *Zfp42* and *Dppa5a*)? It would be interesting indeed to more precisely delineate whether (or not) the latter show a loss of H3K27me3 at their TSS and/or enhancers prior to gene activation. As the present, the authors have very little evidence supporting their statement lines 296-300 that “ for many pluripotency loci the removal of H3K27me3 in the early phase of reprogramming primes them for transcriptional activation in the late phase”. As further evidence, could the authors also corroborate the kinetics of H3K27me3 loss and acquisition of H3K27ac at the two sets of loci? Additionally, it might be interesting to explore whether other genomic and epigenetic features underline the observed difference in the gene activation timing of these sets of loci (e.g. CpG low/intermediate/high TSSs, DNA methylation and H3K9me3 status at TSS and/or enhancers in MEFs). In the same line of thought, could the authors repeat their motif analysis shown in Supplementary Fig. 4g separately for the two sets of loci?

9- Figure 4l - how cluster 1, 2 and 3 correlates with the above sets of loci? Please also show GO analysis for the three clusters in Supplementary Fig. 6i. Line 387, could the authors acknowledge the possibility that JMJD3 might be recruited at loci within cluster 2 by other transcription factors that KLF4 upon reprogramming?

10- Figure 5 – the authors interestingly uncover that JMJD3 form part of an activating protein complexes at enhancers. However, they do not explore whether JMJD3 is essential for the activity of enhancers and the mechanisms

by which JMJD3 mediates its action upon reprogramming and/or in fully reprogrammed cells. Could the authors make use of their rescued JMJD3 KO MEFs with WT JMJD3 and mutated forms to compare the profile of enhancer protein binding and enhancer-promoter interactions in fully reprogrammed clones? Any effort in that direction would be a great addition to this study.

Other minor comments:

1- Line 180 – JMJD3 does not promote reprogramming in a Vc-dependent manner as stated. However, its action seems to be enhanced in the presence of Vc.

2- Figure 5h – there is no indication of the reprogramming timing used in these experiments. Same comment for Supplementary Fig. 8c,e.

3- Supplementary Fig.1j – replicates required as well as statistical analysis.

4- Supplementary Fig.2j – could the authors provide a quantification of GFP+ and NANOG+ colonies in the same experiments?

5- Supplementary Fig. 5b, e and g – triplicate experiments should be best provided.

6- Supplementary Fig. 6h – should the authors confidently use data generated from other reprogramming studies (e.g. Chronis et al.) using different protocols and MEF passage numbers?

7- Supplementary Fig. 7d. Please provide information about datasets used in this

Our responses:

“In this study, Huang et al. ... While this observation is novel and interesting, the authors do not explore whether JMJD3 is essential for enhancer-promoter looping and/or the mechanism by which JMJD3 mediates its action at these sites upon reprogramming and/or in fully reprogrammed cells.”

Response: We thank the reviewer for the positive assessment of our manuscript and the helpful comments. For the last point raised here, please refer to the response to specific comment 10 below.

Please note that we would like to change the title to a new one: “**JMJD3 acts in tandem with KLF4 to facilitate reprogramming to pluripotency**”. We believe this title is more concise and with greater perspective. We hope the reviewer will agree with the change. Of course, if the reviewer and/or editor think it is not appropriate, we can change to: “JMJD3 facilitates the architectural role of KLF4 and enhances mouse somatic cell reprogramming in a senescence dependent manner”.

Specific comments:

1-“*The authors find that ectopically expression JMJD3 similarly enhances Ink4a expression in early and late passage MEFs. However, the starting level of Ink4a expression is higher in the latter. Is this also correlating with higher endogenous Jmjd3 expression in these cells? Please add the expression profiles of Jmjd3 and Utx in Figure 1a.*”

Response: The reviewer raised a relevant point. We checked the mRNA expression of *Jmjd3* and *Utx* during serial passaging of MEFs. From passage 2 to 5, when *Ink4a* expression increases robustly, we did not see any significant change in *Jmjd3* or *Utx*, as measured by RT-qPCR (**REBUTTAL Fig. 7A**) or by RNA-seq (**REBUTTAL Fig. 7B**), indicating that the induction of *Ink4a* with passaging is not related to changes in *Jmjd3* expression. Supporting this, *Ink4a* expression is also controlled by other epigenetic mechanisms such as Polycomb and JHDM1B^{15,16}, and is mainly activated by JMJD3 when the latter is induced upon oncogenic or genotoxic stresses^{17,18}. The RT-qPCR results for *Jmjd3* and *Utx* results are now included in **NEW Figure 1b** and also see page 6 line 136.

2-“*While ectopically expressing JMJD3 reduces the formation of AP+ (early) reprogrammed cells, it favours the formation of GFP+ (fully) reprogrammed cells, implying a role for JMJD3 during late stages of reprogramming. Could this be directly examining using high-resolution cell surface markers and flow cytometry analysis as previously described (O’Malley et al. Nature 2013)?*”

Response: The reviewer gave us an excellent suggestion. We have double stained OG2 MEFs reprogrammed by OSKM with Empty vector or JMJD3 OE at days 5 and 10 using antibodies against CD44 and ICAM1. As shown in **REBUTTAL Figure 8A**, MEFs and ESCs show a distinct staining pattern, as

in the work by O' Malley *et al.*¹⁹. Interestingly, at day 5 of reprogramming, JMJD3 OE delays ICAM1 loss (Q2 to Q3, corresponding to the early stage of reprogramming), whereas at day 10 JMJD3 OE accelerates the appearance of CD44⁺ICAM1⁺ cells (Q1, corresponding to the late stage of reprogramming) (**REBUTTAL Fig. 8A, B**). Our RNA-seq also indicates that ICAM1 is a target of JMJD3 in both MEFs and reprogramming (**REBUTTAL Fig. 8C**). In conclusion, these results further show that JMJD3 OE accelerates the late stage of reprogramming but tends to impair the early stage. This observation fits well with our model that JMJD3 acts as a double-edge sword, simultaneously derailing reprogramming by promoting senescence and pluripotency acquisition. We have now included this information (page 7 line 166) and the relevant reference by O' Malley *et al.* in our revised manuscript (**NEW Supplementary Fig. 1k**).

3-“Figure 2a - different timing of reprogramming (day 5 and day 10) are used in this study. Could the authors also quantify the loss of global H3K27me3 in a catalytic-dependent manner at these two timings to corroborate an early and/or late action of JMJD3? Please note that replicates are required for Figure 2a, c and h as well as statistical analysis. It would also be informative to compare the level of H3K27me3 in pre-IPSCs and fully reprogrammed cells in absence of exogenous JMJD3 to validate whether this epigenetic remodelling is normally occurring during this transition.”

Response: First, regarding the western blot in the earlier Figure 2a, we have substituted it for a new one including day 5 and day 10 reprogramming samples with or without JMJD3 OE and with or without Vc (**NEW Fig. 2a**). These experiments were performed in triplicate (the three experiments are shown in **REBUTTAL Fig. 9A**) but only one is shown in the revised manuscript. Yet, the quantification of the three experiments is shown in a bar graph in **NEW Figure 2a** in the revised manuscript, and in **REBUTTAL Figure 9B**. In these experiments, we could verify the reduction of H3K27me3 by JMJD3 compared to Empty with Vc at both day 5 and day 10. We also noticed that reprogramming cells at day 10 show a higher level of H3K27me3 than day 5. Similarly, we performed new western blots and observed that the global levels of H3K27me3 are higher in iPSCs than in MEFs and gradually increase during reprogramming (**REBUTTAL Fig. 9C, D** and **NEW**

Supplementary Fig. 2a). This information is now mentioned in the revised manuscript page 8 line 201).

Second, we have added statistical analysis to previous Figure 2c and 2h (**NEW Fig. 2c, h**) and shown all replicates in **REBUTTAL Figure 9E-H**.

Third, we have now included pre-iPSCs in the western blots, and the statistical quantification compared with iPSCs and MEFs. There is a moderate reduction in H3K27me3 between pre-iPSCs and iPSCs. These westerns have been performed separately from the whole set of reprogramming time points with or without JMJD3 and Vc for ease of comparison (**REBUTTAL Fig. 9C, D** and **NEW Supplementary Fig. 2a**).

Importantly, all new data support our earlier conclusions.

4-“Figure 2e-g – loss of endogenous JMJD3 expression is shown to inhibit reprogramming in contrast to previous reports (e.g. Zhao W et al. Cell 2013). Does the discrepancy relate to a passage number effect? In other words, would they obtain different results if using late passage (P4) MEFs? Could the authors also show the % of AP+ colonies to evaluate whether JMJD3 knock-down in their system would enhance the formation of early reprogrammed cells, but impedes the formation of fully reprogrammed cells only? In the same line of thought, is JMJD3 required for the reprogramming or the maintenance of fully reprogrammed cells? Ideally, the authors could use a conditional rescued system of JMJD3KO MEFs to further explore the exact stage(s) of reprogramming that is impacted by JMJD3 action and associated effects in terms of chromatin changes and gene expression.”

Response: The reviewer raised relevant points. First, using the new conditional KO (cKO; OG2-*Jmjd3*^{fl/fl}) MEFs (see above response to **item 1 by Reviewer #1**) infected with Cre adeno-associated viruses at P1 (**REBUTTAL Fig. 10A** and **NEW supplementary Fig. 2m**), we monitored the reprogramming efficiency due to *Jmjd3* deficiency in early (P2) and late passage (P4 and P5) MEFs. In contrast to P2 cKO MEFs, P5 cKO MEFs showed an ~2-fold increase in the number of both AP⁺ (added as requested by the reviewer) and GFP⁺ colonies, though the overall number of colonies

was small compared to earlier passages (specially P2) (**REBUTTAL Fig. 10B**). P4 cKO MEFs only showed a trend towards an increase in GFP⁺ colonies but displayed an increase in AP⁺ colonies. We also checked *Ink4a* expression in P2 and P4 (P5 MEFs were too few for the analysis) and found a decrease only in P4 reprogramming MEFs depleted of *Jmjd3* (**REBUTTAL Fig. 10C**), which correlates well with the increase in AP⁺ and GFP⁺ colonies at P4/5. These results confirm that at late passage the effect of senescence is one of the biggest barriers for reprogramming and that in this particular context depleting *Jmjd3* can help improve reprogramming. These new findings further support that the discrepancy between the two studies is related indeed to the effect of cell senescence state. For clarity, we have only included GFP⁺ colony data for P2 and P5 in the revised manuscript (**NEW Supplementary Fig. 2n** and page 10 line 237).

Second, it is known that ESCs display no obvious phenotype in the absence of JMJD3^{9,20}. We have now added this information and mentioned in the revised manuscript (page 4 line 94). This is also consistent with our observation that iPSC generation is not completely abolished when using *Jmjd3* KO MEFs.

Third, we have tried to further pinpoint the exact reprogramming stage(s) impacted by JMJD3 using cKO MEFs and an inducible JMJD3 OE. However, while we could generate the construct, we failed to overexpress it possibly due to the relatively long ORF of JMJD3 (i.e. 1641 a.a.). Nevertheless, although useful, we think that this experiment is not strictly necessary in the context of our entire dataset pointing to the same conclusions, and hope that the reviewer will agree with us.

5-“Figure 3a-c show the enhancement and/or inhibition of gene expression upon ectopic expression or knock-down of JMJD3 in MEFs upon reprogramming. Do these gene expression changes not simply reflect differences in reprogramming efficiency rather than a direct effect of JMJD3 gain-and-loss of function? This needs to be discussed by the authors.”

Response: The reviewer raises an important point. In fact, multiple observations support our model: (1) JMJD3 induces demethylation of the

repressive mark H3K27me3. It is not surprising considering that such effect will likely contribute to gene activation, as we see in **NEW Figure 3a-c** and **Supplementary Figure 3a**. (2) More direct evidence comes from the ChIP-seq data for JMJD3 in previous Figure 4. In this regard, as shown in **REBUTTAL Figure 11**, among the 320 genes that are upregulated by JMJD3 OE in reprogramming day 5 in **NEW Supplementary Figure 3b**, we could identify that 33% are bound by JMJD3 in our ChIP-seq data. This supports that JMJD3 contributes directly to a significant part of the gene expression changes in reprogramming. Nevertheless, to clarify this point we have added the data (**NEW Supplementary Fig. 7h**) and a note in the revised manuscript (page 18 line 475).

6-“Figure 3d – Loss of H3K27me3 seems to happen faster when ectopically expressing JMJD3 in a WT compared to JMJD3KO background. Could the authors discuss this difference? Again, it would be informative to also examine the profile of H3K27me3 in pre-IPSC and fully reprogramming cells (endogenous Jmjd3 action).”

Response: JMJD3 OE reduces H3K27me3 in a WT background compared to Empty control. In the absence of JMJD3, the reduction of H3K27me3 should be less obvious and this is indeed what we observed in previous Figure 3d for *Jmjd3* KO compared to its WT control. We have now explained this clearer in the revised manuscript (page 13 line 324). As for adding pre-iPSCs and iPSCs to this figure, we respectfully think that it would not clarify things further.

7-“Figure 3e – could the authors show the level of H3K27me3 in the same populations at desert regions to access whether the effect is loci specific as opposed to genome-wide?”

Response: The reviewer raised a good point. We have checked the level of H3K27me3 in desert regions and could find a similar pattern as in the genic regions (**REBUTTAL Fig. 12**). This indicates that the effect of JMJD3 is indeed genome-wide. These results are now shown in **NEW Figure 3e** in the revised manuscript (page 13 line 337).

8-“Figure 3f and Supplementary Fig. 4d,e – could the authors separately look at H3K27me3 changes upon reprogramming at the two distinct sets of loci: (1)

loci up-regulated from day 5 (e.g. Cdh1, Sall1 and Sall4) and (2) loci only up-regulated later on at day 10 of reprogramming (e.g. Esrrb, Zfp42 and Dppa5a)? It would be interesting indeed to more precisely delineate whether (or not) the latter show a loss of H3K27me3 at their TSS and/or enhancers prior to gene activation. As the present, the authors have very little evidence supporting their statement lines 296-300 that “for many pluripotency loci the removal of H3K27me3 in the early phase of reprogramming primes them for transcriptional activation in the late phase”. As further evidence, could the authors also corroborate the kinetics of H3K27me3 loss and acquisition of H3K27ac at the two sets of loci? Additionally, it might be interesting to explore whether other genomic and epigenetic features underline the observed difference in the gene activation timing of these sets of loci (e.g. CpG low/intermediate/high TSSs, DNA methylation and H3K9me3 status at TSS and/or enhancers in MEFs). In the same line of thought, could the authors repeat their motif analysis shown in Supplementary Fig. 4g separately for the two sets of loci?”

Response: The reviewer raised relevant points. First, we focused on PSC-enriched differentially expressed genes (DEGs; ‘day 5’ 42 genes, ‘day 10 only’ 100 genes) upregulated upon JMJD3 OE in our own dataset. As shown in **REBUTTAL Figure 13A** upper panel, both ‘day 5’ and ‘day 10 only’ genes showed H3K27me3 demethylation at day 5 in JMJD3 OE at both the TSS and enhancer regions, though this was more prominent for ‘day 5’ genes. We have changed the H3K27me3 panel in previous Figure 3f for a new one separating both groups of genes (**NEW Fig. 3f** upper panels).

Second, H3K27ac showed an increase at day 5 in ‘day 5’ genes upregulated by JMJD3 OE, especially around the TSS. As we proposed, this finding suggests that the H3K27me3-to-H3K27ac switch at day 5 contributes to the activation of these genes (**REBUTTAL Fig. 13A** lower panels and **NEW Fig. 5g**). As for ‘day 10 only’ genes upregulated upon JMJD3 OE, we could see two types of patterns among the well-known pluripotency genes. Genes such as *Mycn*, *Utf1*, *Nanog*, *Tfcp2l1* and *Tdgf1* show different degrees of H3K27me3 demethylation at reprogramming day 5 with JMJD3 OE in either TSS or enhancer regions (**REBUTTAL Fig. 13B** upper panels). Various degrees of H3K27me3-to-H3K27ac switch also happened at some of these

loci like *Mycn*, *Nanog* and *Tfcp2l1*. In contrast, genes such as *Dppa5a*, *Dppa4*, *Zfp42*, *Lin28a* and *Nr5a2*, however, either have no enrichment of H3K27me3 in MEFs or show no demethylation during reprogramming. These genes do not have acquired H3K27ac at day 5 either (**REBUTTAL Fig. 13B** lower panels). The discrepancy among groups of genes is not surprising and indicates that not all pluripotency genes are regulated directly by JMJD3 (see **item 5 by the same reviewer** above). All this is now explained better in the revised text (page 14 line 356 and page 21 line 555) and we have included selected loci in **NEW Supplementary Figure 5a**. In summary, the new analyses support our earlier conclusion that for both early- and late-activated genes, including a significant part of pluripotency genes, JMJD3-mediated H3K27me3 demethylation facilitates an H3K27ac switch and their activation.

Third, we also used the ChIP-seq data from Chronis *et al.*²¹ that profiled MEFs, OSKM reprogramming cells at 48 hours, pre-iPSCs and ESCs. In this dataset, we observed a striking H3K27me3-to-H3K27ac switch but only small changes of H3K9me3 levels from MEFs to ESCs and in early reprogramming (**REBUTTAL Fig. 13C**). Moreover, according to the data from Schwarz *et al.*²²⁻²⁴, the enhancer regions of both 'day 5' and 'day 10 only' genes show a remarkable DNA demethylation from MEFs to ESCs/iPSCs (**REBUTTAL Fig. 13D**), with the latter being more obvious, supporting an involvement of DNA demethylation in their activation. This information is now included in **NEW Figure 3f** and **NEW Supplementary Figure 4d** (page 13 line 349 and page 14 line 368).

Fourth, motif discovery for the enhancers of DEGs that are upregulated upon JMJD3 OE at 'day 5' or at 'day 10 only' shows that OCT family (and also SOX for enhancer loci) transcription factors are enriched in both, whereas KLF family transcription factors are more enriched in the latter (**REBUTTAL Fig. 13E**). Motifs for non-pluripotent transcription factors including for example NKX6.1, SP2/5 and LHX1/2 were also present at both time points. A similar pattern was maintained around the TSS. Because the overall gene number for motif discovery is limited, and not all DEGs are bound by JMJD3, we respectfully do not think it is very informative to include this figure into the revised manuscript.

9-“Figure 4l - how cluster 1, 2 and 3 correlates with the above sets of loci?
Please also show GO analysis for the three clusters in Supplementary Fig. 6i.
Line 387, could the authors acknowledge the possibility that JMJD3 might be recruited at loci within cluster 2 by other transcription factors that KLF4 upon reprogramming?”

Response: The reviewer raised relevant points. First, we overlapped early and late-activated DEGs with the genes in clusters 1, 2 and 3.

(1) KLF4 and/or JMJD3 (all three clusters) bind to 55% of early- and 47% of late-activated DEGs at day 5 (**REBUTTAL Fig. 14A** upper panel). The binding is slightly more evident for PSC-enriched DEGs, with 60% of early- and 55% of late-activated ones (**REBUTTAL Fig. 14A** lower panel). This confirmed that a significant part of transcriptional changes in reprogramming with JMJD3 OE are attributed to the direct binding of KLF4 and/or JMJD3.

(2) KLF4 and JMJD3 (cluster 1) co-bind more early- than late-activated DEGs for both total and PSC-enriched DEGs.

(3) KLF4 (cluster 1 and 3) binds more genes at both stages than JMJD3 (cluster 1 and 2), and in the late stage differences are more striking (46% vs. 33% in early and 42% vs. 22% in late) (**REBUTTAL Fig. 14A** upper panel). The preference for KLF4 was more obvious for PSC-enriched DEGs, with 50% or more genes bound by KLF4 for the two stages.

(4) KLF4 alone (cluster 3) could bind 36% of late-activated PSC-enriched genes in early stage of reprogramming. Upon JMJD3 OE, these genes could be effectively activated in late reprogramming either through an indirect effect or through JMJD3 binding in the late stage.

We have shown these results in **NEW Supplementary Figure 7h** and explained them clearer in the revised text (page 18 line 475).

Second, we have included updated GO analysis for the three clusters in **NEW Supplementary Figure 7f** (see **REBUTTAL Fig. 14B** and page 18 line 468).

Third, motif discovery for cluster 2 in previous Figure 4m indeed suggests that JMJD3 is also recruited by other somatic transcription factors, in particular the AP-1 family. So, we agree with the reviewer on this important

point; we had mentioned this before in the text but now we remarked it (page 17 line 463).

10-“Figure 5 – the authors interestingly uncover that JMJD3 form part of an activating protein complexes at enhancers. However, they do not explore whether JMJD3 is essential for the activity of enhancers and the mechanisms by which JMJD3 mediates its action upon reprogramming and/or in fully reprogrammed cells. Could the authors make use of their rescued JMJD3 KO MEFs with WT JMJD3 and mutated forms to compare the profile of enhancer protein binding and enhancer-promoter interactions in fully reprogrammed clones? Any effort in that direction would be a great addition to this study.”

Response: The reviewer raised interesting points. First, to explore whether endogenous JMJD3 is essential for enhancer activation and enhancer-promoter interactions, we have now knocked down *Jmjd3* in reprogramming and performed ChIP-qPCR to measure the binding of KLF4, NIPBL, SMC1A and the deposition of H3K27ac. Due to the low enrichment for the relevant proteins on the loci of interest in normal reprogramming at day 5 (in part caused by the heterogenous nature of reprogramming), we decided to use day 7 samples instead. As shown in **REBUTTAL Figure 15**, we could see a significant drop for all these proteins in *Jmjd3* knockdown cells. These results are now shown as **NEW Supplementary Figure 8f and 9c** in the revised manuscript (see page 20 line 549 and page 21 line 563).

Second, since JMJD3 deficiency does not block reprogramming completely, and both mouse and human PSCs can be maintained without obvious alterations in the absence of JMJD3^{9,20}, we do not think that the enhancer-promoter looping activity of JMJD3 in reprogramming is indispensable for reprogramming or in fully reprogrammed iPSCs/ESCs. Possibly, JMJD3 activity in this regard is complemented by other factors. We have now mentioned this in the revised manuscript (page 23 line 613). Accordingly, we respectfully do not think that it is necessary to profile enhancer protein binding or enhancer-promoter interactions in iPSCs reprogrammed with or without JMJD3.

Other minor comments:

1-“Line 180 – JMDJ3 does not promote reprogramming in a Vc-dependent manner as stated. However, its action seems to be enhanced in the presence of Vc.”

Response: Thanks for spotting this. Yes, we agree with the reviewer and have revised the text as “...but also promoting reprogramming, which can be further enhanced by Vc” (page 8 line 193).

2-“Figure 5h – there is no indication of the reprogramming timing used in these experiments. Same comment for Supplementary Fig. 8c,e.”

Response: We have amended the labels (including the timing) for these figures. **NEW Figure 5i** is OSKM+Vc D5 and **NEW Supplementary Figure 9e** is OSKM+Vc D10 (this one was already shown in the last submission).

3-“Supplementary Fig.1j – replicates required as well as statistical analysis.”

Response: We have now included two more biological replicates in **REBUTTAL Figure 16A** and a graph bar with the statistical analysis in **NEW Supplementary Figure 1I (REBUTTAL Fig. 16B)**. The results support our previous conclusion that PHF20 is degraded by JMJD3 independently of Vc or cell context.

4-“Supplementary Fig.2j – could the authors provide a quantification of GFP+ and NANOG+ colonies in the same experiments?”

Response: Yes, we have now done three independent repeats and quantified both GFP⁺ and NANOG⁺ colonies in the same experiments (**REBUTTAL Fig. 16C, NEW Supplementary Fig. 2k**, and page 9 line 228). The results support our early conclusion. Moreover, the stringency of OG2 GFP⁺ colony quantification as readout for reprogramming efficiency could be verified using another type of *Dppa5a*-tdTomato/OG2 dual-reporter MEFs²⁵ and measured by flow cytometry (**REBUTTAL Fig. 16D, NEW Supplementary Fig. 1d** and page 7 line 150).

5-“Supplementary Fig. 5b, e and g – triplicate experiments should be best provided.”

Response: Thanks for spotting this. We have now included triplicate experiments in these three figures (**REBUTTAL Fig. 17A-C** and **NEW**

Supplementary Fig. 6b, f, h). The patterns and conclusions are the same as before.

6-“*Supplementary Fig. 6h – should the authors confidently use data generated from other reprogramming studies (e.g. Chronis et al.) using different protocols and MEF passage numbers?*”

Response: We understand the reviewer’s concern. To address this issue, we have now included our own ATAC-seq for OSKM reprogramming cells at two time points, with MEFs and iPSCs as controls. The kinetics of changes for cluster 2 in reprogramming cells compared with MEFs and iPSCs show very similar patterns to those in our previous Supplementary Fig. 6h (**REBUTTAL Fig. 18**). The pattern confirms the transient opening chromatin feature of these genes and highlights consistency between the reprogramming systems in the different datasets. For clarity, we have eliminated previous Supplementary Figure 6f,g and replaced previous Supplementary Figure 6h with a new one (**NEW Supplementary Fig. 7g**).

7-“*Supplementary Fig. 7d. Please provide information about datasets used in this figure.*”

Response: We have now included the accession numbers for all datasets used in our study into the Methods section (see **Data availability/Accession number** and page 35 line 963).

References

1. Zhang, F. *et al.* JMJD3 promotes chondrocyte proliferation and hypertrophy during endochondral bone formation in mice. *J. Mol. Cell Biol.* **7**, 23-34 (2015).
2. Mathieu, J. *et al.* Hypoxia-inducible factors have distinct and stage-specific roles during reprogramming of human cells to pluripotency. *Cell Stem Cell* **14**, 592-605 (2014).
3. Kim, J.B. *et al.* Pluripotent stem cells induced from adult neural stem cells by reprogramming with two factors. *Nature* **454**, 646-650 (2008).
4. Burgold, T. *et al.* The H3K27 demethylase JMJD3 is required for maintenance of the embryonic respiratory neuronal network, neonatal breathing, and survival. *Cell Rep.* **2**, 1244-1258 (2012).
5. Park, D.H. *et al.* Activation of neuronal gene expression by the JMJD3 demethylase is required for postnatal and adult brain neurogenesis. *Cell Rep.* **8**, 1290-1299 (2014).

6. He, X.B. *et al.* Vitamin C facilitates dopamine neuron differentiation in fetal midbrain through TET1- and JMJD3-dependent epigenetic control manner. *Stem Cells* **33**, 1320-1332 (2015).
7. Burchfield, J.S., Li, Q., Wang, H.Y. & Wang, R.F. JMJD3 as an epigenetic regulator in development and disease. *Int. J. Biochem. Cell Biol.* **67**, 148-157 (2015).
8. Burgold, T. *et al.* The histone H3 lysine 27-specific demethylase Jmjd3 is required for neural commitment. *PLoS One* **3**, e3034 (2008).
9. Shan, Y. *et al.* JMJD3 and UTX determine fidelity and lineage specification of human neural progenitor cells. *Nat. Commun.* **11**, 382 (2020).
10. Carey, B.W., Markoulaki, S., Beard, C., Hanna, J. & Jaenisch, R. Single-gene transgenic mouse strains for reprogramming adult somatic cells. *Nat. Methods* **7**, 56-59 (2010).
11. Zhu, S., Wang, H. & Ding, S. Reprogramming fibroblasts toward cardiomyocytes, neural stem cells and hepatocytes by cell activation and signaling-directed lineage conversion. *Nat. Protoc.* **10**, 959-973 (2015).
12. Kim, J. *et al.* Direct reprogramming of mouse fibroblasts to neural progenitors. *Proc. Natl. Acad. Sci. USA* **108**, 7838-7843 (2011).
13. Vierbuchen, T. *et al.* Direct conversion of fibroblasts to functional neurons by defined factors. *Nature* **463**, 1035-1041 (2010).
14. Wray, J. *et al.* Inhibition of glycogen synthase kinase-3 alleviates Tcf3 repression of the pluripotency network and increases embryonic stem cell resistance to differentiation. *Nat. Cell Biol.* **13**, 838-845 (2011).
15. Bracken, A.P. *et al.* The Polycomb group proteins bind throughout the INK4A-ARF locus and are disassociated in senescent cells. *Genes Dev.* **21**, 525-530 (2007).
16. He, J., Kallin, E.M., Tsukada, Y. & Zhang, Y. The H3K36 demethylase Jhdm1b/Kdm2b regulates cell proliferation and senescence through p15(Ink4b). *Nat. Struct. Mol. Biol.* **15**, 1169-1175 (2008).
17. Agger, K. *et al.* The H3K27me3 demethylase JMJD3 contributes to the activation of the INK4A-ARF locus in response to oncogene- and stress-induced senescence. *Genes Dev.* **23**, 1171-1176 (2009).
18. Barradas, M. *et al.* Histone demethylase JMJD3 contributes to epigenetic control of INK4a/ARF by oncogenic RAS. *Genes Dev.* **23**, 1177-1182 (2009).
19. O'Malley, J. *et al.* High-resolution analysis with novel cell-surface markers identifies routes to iPS cells. *Nature* **499**, 88-91 (2013).
20. Ohtani, K. *et al.* Jmjd3 controls mesodermal and cardiovascular differentiation of embryonic stem cells. *Circ. Res.* **113**, 856-862 (2013).

21. Chronis, C. *et al.* Cooperative Binding of Transcription Factors Orchestrates Reprogramming. *Cell* **168**, 442-459 e420 (2017).
22. Schwarz, B.A. *et al.* Prospective Isolation of Poised iPSC Intermediates Reveals Principles of Cellular Reprogramming. *Cell Stem Cell* **23**, 289-305 e285 (2018).
23. Liu, Y. *et al.* Bisulfite-free direct detection of 5-methylcytosine and 5-hydroxymethylcytosine at base resolution. *Nat. Biotechnol.* **37**, 424-429 (2019).
24. Lu, F., Liu, Y., Jiang, L., Yamaguchi, S. & Zhang, Y. Role of Tet proteins in enhancer activity and telomere elongation. *Genes Dev.* **28**, 2103-2119 (2014).
25. Guo, L. *et al.* Resolving Cell Fate Decisions during Somatic Cell Reprogramming by Single-Cell RNA-Seq. *Mol. Cell* **73**, 815-829 e817 (2019).

REVIEWERS' COMMENTS:

Reviewer #1 (Remarks to the Author):

The authors have addressed most of my concerns. I still would like to see the raw FACS data in "NEW Supplementary Fig. 6i", instead of the current plot.

Reviewer #3 (Remarks to the Author):

I have now reviewed the rebuttal letter and revised manuscript by Jinhua Huang et al. now entitled "Jmjd3 acts in tandem with KLF4 to facilitate reprogramming to pluripotency". I would like to thank the authors for all their efforts and most adequate replies to my comments and suggestions. Most importantly, the authors have addressed the technical concerns I had raised at the first place. I have not further comments to submit to the authors.

REPLY TO REVIEWERS' COMMENTS:

Reviewer #1:

"The authors have addressed most of my concerns. I still would like to see the raw FACS data in "NEW Supplementary Fig. 6i", instead of the current plot."

Response: We thank the reviewer for the positive assessment of our manuscript and the remaining suggestion. We have presented the raw FACS data of Epi-to-naïve transition for Rex1GFPd2 EpiLCs in **NEW Fig. 4h**, and a larger figure showing the sequential gating strategy in **NEW Supplementary Fig. 13**.

Reviewer #3 (Remarks to the Author):

"I have now reviewed the rebuttal letter and revised manuscript by Jinhua Huang et al. now entitled "Jmjd3 acts in tandem with KLF4 to facilitate reprogramming to pluripotency". I would like to thank the authors for all their efforts and most adequate replies to my comments and suggestions. Most importantly, the authors have addressed the technical concerns I had raised at the first place. I have not further comments to submit to the authors."

Response: We thank the reviewer for the positive assessment of our manuscript.